# An Efficient Sparse Fine-Tuning with Low Quantization Error via Neural Network Pruning

**Cen-Jhih Li**  *cenjhih.li@gmail.com*
*Kahlert School of Computing*
*University of Utah, Salt Lake City, UT 84112, USA*

**Aditya Bhaskara**  *bhaskaraaditya@gmail.com*
*Kahlert School of Computing*
*University of Utah, Salt Lake City, UT 84112, USA*

**Reviewed on OpenReview:** *https://openreview.net/forum?id=w3b67v5EzD*

## Abstract

Fine-tuning is an important step in adapting foundation models such as large language models to downstream tasks. To make this step more accessible to users with limited computational budgets, it is crucial to develop fine-tuning methods that are memory and computationally efficient. Sparse Fine-tuning (SpFT) and Low-rank adaptation (LoRA) are two frameworks that have emerged for addressing this problem and have been adopted widely in practice. In this work, we develop a new SpFT framework, based on ideas from neural network pruning. At a high level, we first identify "important" neurons/nodes using feature importance metrics from network pruning (specifically, we use the structural pruning method), and then perform fine-tuning by restricting to weights involving these neurons. Experiments on common language tasks show our method improves SpFT's memory efficiency by 20-50% while matching the accuracy of state-of-the-art methods like LoRA's variants. Code available at: `https://github.com/CenjhihLi/sparsity_finetuning`.

## 1 Introduction

The paradigm of *pre-training followed by fine-tuning* has seen tremendous success in the last few years. Very large models (often referred to as foundation models) are first trained, typically using very large amounts of data and computational resources, using self-supervised learning approaches (Dosovitskiy, 2020; Achiam et al., 2023; Dubey et al., 2024; Zhou et al., 2024). When building a model for a new task (which could be a supervised learning task), the idea is to start with the foundation model and then tune its parameters, possibly after adding additional classification layers, by training using task-specific data. The pre-train then fine-tune paradigm has been shown to have significant advantages over training a new model from scratch for the new task. Often, high accuracy can be obtained using much smaller datasets for the new task.

Despite the success, fine-tuning a model with billions of parameters requires access to heavy computational resources, even when the task datasets are fairly small. Fortunately, studies (e.g., Panigrahi et al. (2023) and references therein) show that fine-tuning only a small fraction of parameters can be effective. Parameter-efficient fine-tuning (PEFT) methods have thus been proposed to carry out this idea and address the challenge of making fine-tuning more accessible (Lialin et al., 2023). A leading PEFT approach, Low-Rank Adaptation (LoRA, Hu et al. 2022), achieves memory efficiency by simply making low-rank updates to the weight matrices in the different layers. Another class of PEFT methods is *sparse* fine-tuning (SpFT, Sung et al. 2021; Guo et al. 2021; Ansell et al. 2022; Nikdan et al. 2024; Guo et al. 2024b; Panda et al. 2024), which learns a sparse matrix, typically an unstructured one, for updating the pre-trained weights. However, SpFT typically incurs higher memory costs than LoRA during the fine-tuning process, because of the unstructured sparsity. Several works aim to mitigate the memory complexity of SpFT (Mofrad et al., 2019; Holmes et al., 2021; Nikdan

et al., 2023; 2024), often at the cost of increased running time and more complex implementations of sparse kernels. Besides PEFTs, techniques like Zeroth-Order optimization (Malladi et al., 2023; Guo et al., 2024c) and quantization (Gholami et al., 2022; Dettmers et al., 2022; 2024) can further enhance memory and training efficiency for fine-tuning, including LoRA and SpFT.

As LLMs increase in scale, advancing efficient sparse matrix computation, PEFT, and efficient training remains a crucial problem. Towards this goal, we study the question: *Can sparse fine-tuning be improved to create a memory- and parameter-efficient framework, while avoiding additional implementations of sparse operations and without increasing the training time complexity?* We answer this question in the affirmative, by proposing a new SpFT framework for fine-tuning LLMs that achieves memory- and parameter-efficiency while maintaining or even improving performance on downstream tasks. Our approach utilizes NN pruning techniques to identify a subset of fine-tuning parameters and employs a matrix decomposition-based computation for efficient fine-tuning. This design enables the integration of ideas from model compression, SpFT, and matrix decomposition methods.

## 1.1 Our Contributions

At a high level, our contributions are as follows:

- We leverage ideas from *network pruning* to improve SpFT, achieving a significant memory efficiency considerably lower than that of the popular LoRA. Our method uses only standard tensor operations, eliminating the need for custom sparse tensor operations. Additionally, our approach supports fine-tuning quantized base models to further reduce memory footprints.

- We analyze the memory assignment of several PEFT methods and suggest that *computation graphs can affect memory more significantly* than the number of trainable parameters. In addition, we validate our methods in various fine-tuning tasks and provide practical guidance on training strategies.

- We propose two variants of the Taylor importance for different settings in image and language tasks: *class-aware* Taylor and Zeroth-Order Taylor. The first is designed for tasks where class-wise accuracy is important (in addition to overall accuracy), such as image classification. Zeroth-Order Taylor is designed for large language models and requires memory only *equal to that of a forward pass.* In addition, we show how to effectively reduce the estimation variance of the Zeroth-Order estimator.

The rest of the paper is organized as follows. We discuss existing PEFT methods in Section 2 and analyze the existing problem in memory efficiency in Section 3. Following this, we describe our approach in detail in Section 4. Section 5 describes the settings of our experiments. We then present and discuss our results in Section 6. Section 7 concludes with some directions for future work along our lines.

## 2 Background and Related Work

**Parameter-Efficient and Memory-Efficient Fine-Tuning**: In various language and vision tasks, the "pre-train then fine-tune" paradigm has been shown highly effective. PEFT methods (Lialin et al., 2023) fine-tune a small subset of the parameters of a large pre-trained model in order to accelerate the training process. We begin by introducing SpFT and LoRA, two popular approaches for PEFT.

**Sparse Fine-Tuning**: SpFT formulates the fine-tuning process as learning another weight matrix $\boldsymbol{\delta}$:

$$\hat{\mathbf{W}} = \mathbf{W} + \boldsymbol{\delta}, \tag{1}$$

$$\mathbf{h} = f(\hat{\mathbf{W}}, \mathbf{x}) = f(\mathbf{W} + \boldsymbol{\delta}, \mathbf{x}), \tag{2}$$

where $\mathbf{h} \in \mathbb{R}^{d_{out}}$ and $\mathbf{x} \in \mathbb{R}^{d_{in}}$ are the input and output of the hidden layer, respectively, $f(\cdot)$ is the forward function, $\mathbf{W} \in \mathbb{R}^{d_{out} \times d_{in}}$ represents the frozen pre-trained parameters, and $\hat{\mathbf{W}} \in \mathbb{R}^{d_{out} \times d_{in}}$ denotes the final parameters used during inference for the fine-tuning task. The matrix $\boldsymbol{\delta} \in \mathbb{R}^{d_{out} \times d_{in}}$ is sparse and is initialized at $\mathbf{0}$. Such a decomposition is done for every layer in the neural network. SpFT methods try

to limit the number of parameters to fine-tune. For selecting non-zero indices, *Diff pruning* (Guo et al., 2021) learns a mask for $\boldsymbol{\delta}$ (using a standard Backprop algorithm), while *FISH Mask* (Sung et al., 2021) uses Fisher information to identify important indices in $\mathbf{W}$. *Lottery Ticket SpFT* (Ansell et al., 2022) fine-tune the whole model for one epoch, then use $\boldsymbol{\delta}$ itself as an importance score to decide which parameters to fine-tune subsequently. *Robust Adaptor* (RoSA, Nikdan et al. 2024) combines the above SpFTs with LoRA and outperforms all these approaches. However, the key challenge of all SpFT methods is that they do not sufficiently reduce memory usage, as $\boldsymbol{\delta}$ keeps the dimensionality of $\mathbf{W}$, and thus standard libraries do not yield memory improvements.

**Techniques for Efficient Sparse Computation**: To reduce memory redundancy in sparse tensor computations, various data formats like compressed sparse column/row (CSC/CSR, Mofrad et al., 2019; Lu et al., 2024) and semi-structured formats (Holmes et al., 2021) have been proposed. These formats enable efficient operations like Sparse Matrix Multiplication (SpMM), which is crucial for dot products and matrix multiplications. Upon these techniques, sparse backpropagation is built to improve training efficiency (Zhang et al., 2020; Gale et al., 2020; Peste et al., 2021; Schwarz et al., 2021; Hoefler et al., 2021; Jiang et al., 2022; Nikdan et al., 2023; Xu et al., 2024). Beyond sparse tensor techniques, NVIDIA also offers memory optimization techniques for efficient training[1].

However, these techniques come with trade-offs, particularly in terms of time complexity and implementation complexity. Achieving memory efficiency often requires a significant increase in time complexity. To mitigate this, some approaches employ optimizations implemented in C++ or lower-level languages, such as those used in (Gale et al., 2020; Nikdan et al., 2023; 2024), to accelerate the training process.

**Low-Rank Adaptation (LoRA)**: Instead of requiring $\boldsymbol{\delta}$ to be sparse, low-rank adaptation aims to find update matrices that are of small rank:

$$\hat{\mathbf{W}} = \mathbf{W} + \frac{\alpha}{r}\mathbf{B}\mathbf{A}, \tag{3}$$

$$\mathbf{h} = f(\hat{\mathbf{W}}, \mathbf{x}) = f(\mathbf{W}, \mathbf{x}) + f(\frac{\alpha}{r}\mathbf{B}\mathbf{A}, \mathbf{x}), \tag{4}$$

where $\alpha$ is the LoRA scaling hyper-parameter, $\mathbf{B} \in \mathbb{R}^{d_{out} \times r}$, $\mathbf{A} \in \mathbb{R}^{r \times d_{in}}$ are the low-rank matrices with $r \ll d_{in}, d_{out}$. During inference, the $\mathbf{B}\mathbf{A}$ term can be merged into $\mathbf{W}$ to maintain the inference latency of the original model. During training, owing to the fact that $f$ is additive for both the self-attention blocks and the subsequent feed-forwarding neworks (FFN) of transformers (Vaswani, 2017), backpropagation can be performed efficiently for the $\mathbf{B}, \mathbf{A}$ parameters. Due to LoRA's simplicity and effectiveness, numerous variants have been proposed to enhance the performance, e.g., QLoRA (Dettmers et al., 2022; Guo et al., 2024a; Li et al., 2024; Dettmers et al., 2024), DoRA (Liu et al., 2024), RoSA (Nikdan et al., 2024), VeRA (Kopiczko et al., 2024), and S-LoRA (Sheng et al., 2023). These methods have achieved exceptional performance, often comparable to full fine-tuning across a range of tasks.

**Neural Network Pruning**: Besides PEFTs, neural network pruning is another widely applied technique that exploits parameter sparsity to reduce model complexity and speed up inference (LeCun et al., 1989; Han et al., 2015; Han, 2017; Hoefler et al., 2021). Most pruning methods assess *importance* of neural network weights (or neurons) and remove the least important parameters. *Unstructured* pruning zeros out individual weights while preserving the network architecture, whereas *structured* pruning removes parameter groups like channels or neurons, which reduce model size (Liu et al., 2021; Fang et al., 2023; Ma et al., 2023). Both approaches often require retraining to recover lost accuracy during pruning. While effective for classical NNs, pruning LLMs is costly due to high memory demands for computing importance scores and the prohibitive retraining step, making memory-efficient LLM pruning an active research area (Frantar & Alistarh, 2023; Sun et al., 2024).

**Differentiate Our Work From Prior Works**: Although prior SpFT and neural network pruning studies have explored training only a subset of model parameters, they rarely highlight the substantial memory overhead and practical difficulties of unstructured sparsity. In contrast, our method uses structured row selection, enabling sparse fine-tuning with standard dense tensor operations. Furthermore, by selecting

---

[1]Available at `https://pytorch.org/torchtune/stable/tutorials/memory_optimizations.html`

entire rows, our approach guarantees zero quantization error on the fine-tuned rows in quantized models, whereas SVD-based techniques can only reduce, not eliminate, such error. In addition, to our knowledge, the Zeroth-Order Taylor method is the first to leverage zero-order estimation for computing neuron importance scores, with theoretical analysis supporting its effectiveness.

## 3    Number of Trainable Parameters Is Not Everything

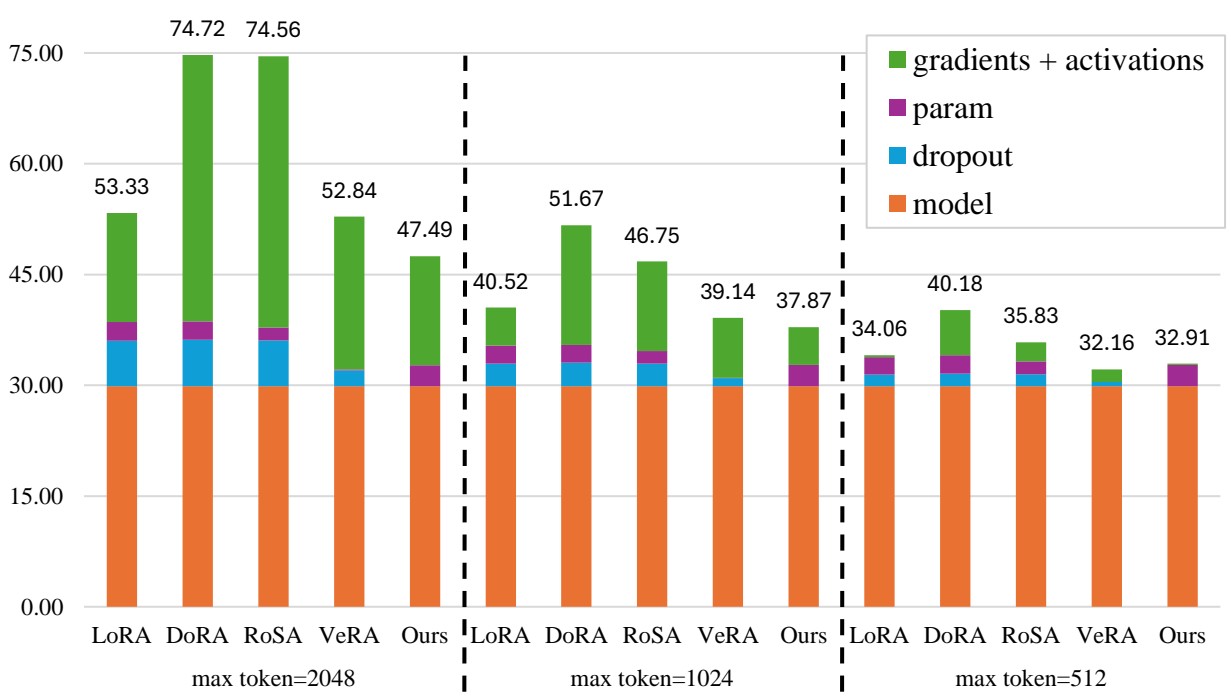

Figure 1: Memory footprints (in GB) of fine-tuning full precision Llama2 using LoRA (Hu et al., 2022), RoSA (Nikdan et al., 2024), DoRA (Liu et al., 2024), VeRA (Kopiczko et al., 2024), and ours. We set $r = 32, d = 1.2\%$ for RoSA, $r = 128$ for ours, and $r = 64$ for the others. Note that RoSA has its own official implementation, which may influence memory consumption, whereas LoRA, DoRA, and VeRA are integrated into the PEFT library provided by Hugging Face. More details please see Appendix D.3.

Before introducing our approach, we want to emphasize that *in PEFT research, reducing the number of trainable parameters is not the most critical factor for minimizing memory consumption.* While certain PEFT methods explicitly aim to lower the number of trainable parameters to reduce memory usage, the impact of this reduction diminishes once the parameter count is sufficiently small. To investigate this further, we compare several representative methods of SpFT and low-rank methods, focusing on their memory footprints during training, as shown in Figure 1. For precisely, we assume full-precision training to exclude memory costs introduced by quantization or other compression techniques and implementations. Notably, quantizing the model to 4-bit precision can yield an additional memory savings of approximately 75%.

During neural network training, backpropagation requires caching a large number of intermediate activations to compute gradients efficiently. The memory cost of these intermediate values is largely influenced by the structure of the computation graph. When the number of trainable parameters is small, the memory consumed by intermediate activations (see green bars in Figure 1) often dominates memory usage apart from the model weights.

Among the LoRA-based methods, VeRA attempts to reduce memory by sharing a pair of low-rank matrices across layers, thereby reducing the number of trainable parameters. However, as shown in Figure 1, this results

in only marginal memory savings—around 0.5GB to 2GB depending on the maximum token length, which is almost negligible. In contrast, DoRA and RoSA incur significantly higher memory usage due to their more complex computation graphs and reliance on unstructured sparse matrices. For instance, DoRA decomposes LoRA's matrices into separate magnitude and direction components (see Figure 4 in Appendix D.4), which substantially increases memory requirements for activation caching. While DoRA's trainable parameter cost is similar to that of LoRA, its overall memory consumption is considerably higher. RoSA also consumes much more memory than LoRA despite incorporating efficiency-oriented design choices. These findings suggest that a simple computation graph can be a far more significant contributor to memory usage than reducing trainable parameters.

## 4 Our Method

To address the challenges mentioned above, we propose **S**tructured-**Pru**ning-based Sparse **F**ine-**T**uning (SPruFT), as illustrated in Figure 2. This is a novel approach designed to streamline computation graphs and eliminate the need for implementing sparse tensor operations. This method ensures memory efficiency while maintaining competitive fine-tuning performance.

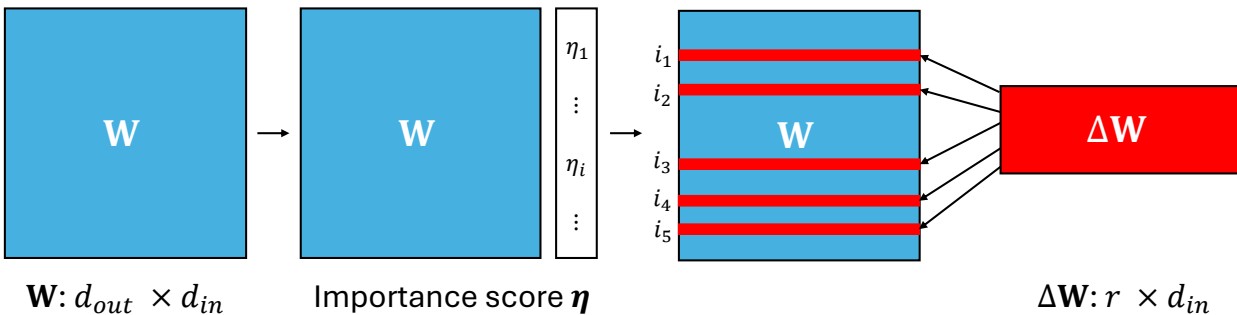

Figure 2: The illustration of SPruFT: we evaluate the importance score for each neuron to select the fine-tuning indices. Then we construct the lower-dimensional fine-tuning parameter matrix $\Delta \mathbf{W}$.

### 4.1 Proposed Method

SPruFT utilizes structured neural network pruning to select a subset of the parameters for fine-tuning. NN pruning methods have been studied extensively (see Section 2) with the goal of reducing the size of a network (often fewer neurons) while preserving accuracy. These methods develop techniques for identifying neurons that are *important* for a given task. Our key insight is to use these importance metrics to indicate which neurons to focus on during fine-tuning. Note that, unlike pruning, where importance often reflects a neuron's role in the *original* task, here it pertains to the downstream *fine-tuning* task, which may have a different input distribution and loss. In Section 4.3, we discuss various importance metrics from pruning research and discuss their use in fine-tuning.

Our method selects the top-$r$ important neurons based on an importance score $\boldsymbol{\eta}$, where $r$ is determined by the desired number of fine-tuning parameters. It follows that the choice of importance matric becomes crucial, which we discuss in Section 4.3. Let the top $r$ neuron indices be $i_1, i_2, \ldots, i_r$. After obtaining $\boldsymbol{\eta}$, we next construct a lower-dimensional parameter matrix $\Delta \mathbf{W} \in \mathbb{R}^{r \times d_{in}}$, with the row selection matrix $\mathbf{M}_{i_j j} = 1$ for all $j \in [r]$ and zeros elsewhere, where $\mathbf{M}$ can be viewed as a mask. Using notations from Section 2, we initialize $\Delta \mathbf{W}$ to zero and define the final parameters $\hat{\mathbf{W}}$ as Equation 1 where $\boldsymbol{\delta} = \mathbf{M}\Delta \mathbf{W}$.

Let us now examine how to implement the forward to make backpropagation memory-efficient[2]. If the computation graph were to pass through $\mathbf{W} + \mathbf{M}\Delta \mathbf{W}$ (as a naïve implementation would), the gradients would be computed for all $d_{in} \times d_{out}$ parameters, which is redundant. Instead, we use the additivity of the

---

[2]While updating the corresponding rows of $\mathbf{W}$ is the most efficient training, updating $\boldsymbol{\delta}$ provides more flexibility for adapting multiple tasks, see discussion in LoRA (Hu et al., 2022).

forward function: we have, analogous to the setting of LoRA,

$$f(\hat{\mathbf{W}}, \mathbf{x}) = f(\mathbf{W} + \mathbf{M}\Delta\mathbf{W}, \mathbf{x}) = f(\mathbf{W}, \mathbf{x}) + f(\mathbf{M}\Delta\mathbf{W}, \mathbf{x}), \tag{5}$$

As $\mathbf{W}$ remains frozen during fine-tuning, backpropagation only needs to keep track of the derivatives of the second term on the RHS. In addition, $\mathbf{M}$ is now a *fixed* matrix, so the only trainable parameters are those in $\Delta\mathbf{W}$ and $f(\Delta\mathbf{W}, \mathbf{x})$ will not be cached, while LoRA requires the cache of $f(\mathbf{A}, \mathbf{x})$ for computing $\frac{\partial \mathbf{h}}{\partial \mathbf{B}}$ (backpropagation, Rumelhart et al. 1986). Besides, as an SpFT framework, our method does not rely on any dropout layer, which also saves a huge amount of memory. We explain this in detail in Appendix D.5 and we show that the benefits in terms of memory cost are significant in Section 6.1.

## 4.2 QSPruFT: Extension Approach with Low Quantization Error

Our approach is model-agnostic, allowing users to integrate recent advances in PEFT such as QLoRA (Dettmers et al., 2024), LoftQ (Li et al., 2024), and QPiSSA (Meng et al., 2024). QLoRA quantizes the base model to Normal Float 4-bit (NF4) and fine-tunes full-precision LoRA matrices $\mathbf{A}$ and $\mathbf{B}$. Based on QLoRA, LoftQ and QPiSSA propose alternative initialization strategies for $\mathbf{A}$ and $\mathbf{B}$ to reduce quantization error. This error is defined as:

$$\mathbf{W}^{res} = \mathbf{W} - \mathbf{W}^{\text{NF4}}, \tag{6}$$

and both LoftQ and QPiSSA use singular value decomposition (SVD) to initialize $\mathbf{A}$ and $\mathbf{B}$ such that $\mathbf{B}^{init}\mathbf{A}^{init} \approx \mathbf{W}^{res}$.

In our method, when applied to quantized base models, we do not require decomposition to approximate $\mathbf{W}^{res}$. Since we fine-tune only selected rows, we can directly initialize $\Delta\mathbf{W}$ using the corresponding rows from $\mathbf{W}^{res}$, where *the quantization error of the fine-tuning rows is **zero***. This offers a potential advantage over QLoRA, QDoRA, LoftQ, and QPiSSA in terms of accuracy.

## 4.3 Broader Impacts: Importance Metrics

Importance evaluation plays a crucial role in our approach, as discussed above. We try various choices in our work: the first is the simple $\ell_2$ norm of the weight vector corresponding to each neuron; the second is the widely-used Taylor importance (LeCun et al., 1989). By considering the gradients, Taylor importance captures more information about the input distribution as well as the relevance of a neuron for the fine-tuning task of interest (which can be different from the original model). We also consider different variants of Taylor importance, as we discuss below. We remark that norm-based importance can be quite powerful on its own, as is the case with norm-sampling in the matrix approximation literature (Frieze et al., 2004).

### 4.3.1 Class Aware Taylor Importance

In our experiments on image classification tasks, we also consider a "class aware" variant of Taylor importance, which may be of independent interest. The motivation here comes from the observation that the importance of a neuron may depend on the class of an input example (as a toy example, a whisker-detecting neuron may be very important to the cat class, but not much to others; hence not too important on average). Another motivation comes from the observation that when we perform a vanilla (class agnostic) fine-tuning, the accuracy of some classes can be much worse than others — an undesirable outcome. This is shown in Table 1.

We define the class-wise Taylor importance as follows: for neuron $i$ and label $t$,

$$\boldsymbol{\eta}_i^t := |L(\mathcal{D}^t, \acute{F}_i) - L(\mathcal{D}^t, F)| \approx |(\boldsymbol{\theta}^{(i)})^\top \nabla_{\boldsymbol{\theta}^{(i)}} L(\mathcal{D}^t, F)|, \tag{7}$$

where $F$ is the forward function of the entire model, $L(\mathcal{D}^t, F)$ denotes the average loss of $F$ over inputs in class $t$, $\acute{F}_i$ represents the forward without channel/neuron $i$, and $\boldsymbol{\theta}^{(i)}$ is the parameter vector of channel $i$. One natural choice of importance of neuron $i$ motivated by the above discussion is $\max_t \boldsymbol{\eta}_i^t$. We find that this score is "too sensitive" (importance of neurons may be over-estimated because of just one class), leading to lower overall accuracy. On the other hand, the average (over $t$) of $\eta_i^t$ corresponds to the standard

Table 1: The distribution of accuracies across different labels is summarized by statistics including the minimum (Min), first quartile (Q1), median (Med), third quartile (Q3), and maximum (Max) accuracies. #labels is the number of labels. The reported accuracies are the validation results of full fine-tuning DeiT for 5 epochs. Models and Datasets are described in Section 5.

|  | #labels | Mean | Min | Q1 | Med | Q3 | Max |
|---|---|---|---|---|---|---|---|
| CIFAR100 | 100 | 90.18 | 65 | 88 | 92 | 95 | 99 |
| Tiny-ImageNet | 200 | 87.55 | 62 | 84 | 88 | 92 | 100 |

Taylor importance. We find that the intermediate quantity of *Quantiles-Mean*, defined as the average of the $0\%, 10\%, 20\%, \ldots, 100\%$ quantiles of the $\boldsymbol{\eta}_i^t$, works well in reducing the accuracy imbalance across labels, and also achieving a high overall accuracy. Formally,

$$\boldsymbol{\eta}_i = \frac{\sum_{l=0}^{10} Q_l(\{\boldsymbol{\eta}_i{}^t\}_{t=1}^p)}{11},$$

(8)

where $Q_l$ represents the $l \times 10$-th quantile. See Appendix A for more details.

### 4.3.2 Zero-Order Taylor Importance

As discussed, Taylor importance can incorporate information about the data distribution and the fine-tuning task when evaluating important neurons. However, for large models like Llama-3, it turns out that the computational overhead required for computing Taylor importances is prohibitively large.[3] In these cases, we apply the idea from the memory-efficient zeroth-order optimizer (MeZO, Malladi et al. 2023) to estimate the gradient in Taylor importance. The classical Zeroth-Order (ZO) setting is defined as below.

**Definition 4.1** (Simultaneous Perturbation Stochastic Approximation or SPSA (Spall, 1992)). Given a model with parameters $\boldsymbol{\theta} \in \mathbb{R}^d$ and a loss function $L$, SPSA estimates the gradient on a dataset $\mathcal{D}$ as

$$\hat{\nabla} L(\boldsymbol{\theta}, \mathcal{D}) = \frac{L(\boldsymbol{\theta} + \epsilon \mathbf{z}) - L(\boldsymbol{\theta} - \epsilon \mathbf{z})}{2\epsilon} \mathbf{z},$$

where $\mathbf{z} \in \mathbb{R}^d$ is drawn from $\mathbf{z} \sim \mathcal{N}(0, \mathbf{I}_d)$, and $\epsilon$ is the scale of the perturbation. The $n$-SPSA estimate averages $\hat{\nabla} L(\boldsymbol{\theta}, \mathcal{D})$ over $n$ randomly sampled $\mathbf{z}$. Note that as $\epsilon \to 0$, the estimate above converges to $\mathbf{z}(\mathbf{z}^\top \nabla L(\boldsymbol{\theta}, \mathcal{D}))$, which is equal to $\nabla L(\boldsymbol{\theta}, \mathcal{D})$ in expectation.

By applying SPSA, the Zero-Order Taylor (ZOTaylor) importance can be defined as follows:

$$\boldsymbol{\eta}_i := |(\boldsymbol{\theta}^{(i)})^\top \mathbf{g}^{(i)}|, \hat{\boldsymbol{\eta}}_i := |(\boldsymbol{\theta}^{(i)})^\top \hat{\mathbf{g}}^{(i)}|,$$

(9)

where we denote $[\nabla L(\boldsymbol{\theta}, \mathcal{D})]$ and its estimate as $\mathbf{g}$ and $\hat{\mathbf{g}}$ for convenience, and $\boldsymbol{\eta}_i$, $(\boldsymbol{\theta}^{(i)})^\top$, and $\mathbf{g}^{(i)}$ are the importance score, the parameter vector, and the gradient vector for neuron $i$.

We now assess the efficiency and effectiveness of ZOTaylor for our LLM. For efficiency, a naïve implementation of SPSA still requires twice the memory of inference because of the need to store $\mathbf{z}$. However, MeZO uses the trick of regenerating $\mathbf{z}$ dynamically using a random seed (of much smaller size than the model), thus eliminating the storage and ensuring memory usage that is equal to that of inference. We then justify the effectiveness of ZOTaylor in the following.

**Property 4.2.** *$n$-SPSA is an unbiased and consistent estimator with the variance vector $(\sigma_1^2, \cdots, \sigma_d^2)$ where*

$$\sigma_j^2 = \frac{\mathbf{g}_j^2 + \sum_{l=1}^d \mathbf{g}_l^2}{n}.$$

Property 4.2 can be proved by simply noting that the covariance matrix of $\mathbf{z}$ is $\mathbf{I}_d$, details can be found in Appendix B. With this property, we know that $n$-SPSA can accurately estimate the gradient. However,

---

[3]The Taylor importance here refers to computing the exact value without relying on approximations of the importance score or the gradient matrix used for deriving the importance score.

the variance can be large when $d$ is large. Here, we first note that since we aim to find the most important neurons, we do not care about the gradient estimate itself. That is, given $\boldsymbol{\eta}_{i_1} > \boldsymbol{\eta}_{i_2}$, our goal is to have a higher probability $p\left(\hat{\boldsymbol{\eta}}_{i_1} - \hat{\boldsymbol{\eta}}_{i_2} > 0\right)$ which is exactly equal to:

$$p_{Z \sim \mathcal{N}(0,1)} \left( Z > -\frac{|\mathbf{g}^{(i_1)} \boldsymbol{\theta}^{(i_1)}| - |\mathbf{g}^{(i_2)} \boldsymbol{\theta}^{(i_2)}|}{\sqrt{\left(\frac{\sum_{j \in \{j^{(i_1)}\}} \sigma_j^2 \boldsymbol{\theta}_j^2}{2} + \frac{\sum_{j \in \{j^{(i_2)}\}} \sigma_j^2 \boldsymbol{\theta}_j^2}{2}\right)/2}} \right), \qquad (10)$$

where $\{j^{(i_1)}\}$ and $\{j^{(i_2)}\}$ are the indices of neuron $i_1$ and $i_2$, and $d_{i_1} = |\{j^{(i_1)}\}|, d_{i_2} = |\{j^{(i_2)}\}|$. The trivial lower bound of Equation 10 is 0.5 and the probability will close to 0.5 when the variance is large or when $\boldsymbol{\eta}_{i_1}$ and $\boldsymbol{\eta}_{i_2}$ are too close to one another. In our experiments, the values of $\hat{\mathbf{g}}_j$ from a single SPSA are almost uniformly in $[-100, 100]$ with variances ranging from $10^7$ to $10^8$ and the values of $\boldsymbol{\theta}_j$ are mostly in $[-1, 1]$, thus the probability above turns out to be too close to 0.5.

To address this issue, we utilize a simple but highly effective strategy (J Reddi et al., 2015): we partition the training data into $k$ calibration sets. For each calibration set, we generate $n$ distinct perturbations $\mathbf{z}$ to perform $n$-SPSA, producing $n \times k$ gradient estimates. Consequently, the variance of $n$-SPSA is effectively reduced to $\frac{\mathbf{g}_j^2 + \sum_{l=1}^d \mathbf{g}_l^2}{nk}$[4].

**Theorem 4.3.** *Suppose a model parameter $\boldsymbol{\theta}$ satisfies $|\boldsymbol{\theta}|_\infty \leq u_{\boldsymbol{\theta}}$ and a gradient $\mathbf{g}$ satisfies $|\mathbf{g}|_2 \leq u_{\mathbf{g}}$ for some parameter $u_{\mathbf{g}}$. Also assume that $\boldsymbol{\eta}_i$'s are drawn IID from a distribution $\mathcal{P}_{\boldsymbol{\eta}}$ that is $\alpha$-smooth and supported in the interval $[0, u_{\boldsymbol{\eta}}]$. Let $\boldsymbol{\eta}_{i_1}$ and $\boldsymbol{\eta}_{i_2}$ be the importance values of the top-$q_1\%$ and top-$q_2\%$ important neurons and define $\Phi$ to be the CDF of Gaussian distribution, then we have*

$$p\left(\hat{\boldsymbol{\eta}}_{i_1} - \hat{\boldsymbol{\eta}}_{i_2} > 0\right) \geq p_{Z \sim \mathcal{N}(0,1)} \left( Z > -\frac{2\sqrt{nk}(\boldsymbol{\eta}_{i_1} - \boldsymbol{\eta}_{i_2})}{\sqrt{(d_{i_1} + d_{i_2})u_{\boldsymbol{\theta}}^2 u_{\mathbf{g}}^2}} \right) \geq 1 - \Phi\left( -\frac{2\sqrt{nk}(q_2 - q_1)/100}{\alpha\sqrt{(d_{i_1} + d_{i_2})u_{\boldsymbol{\theta}}^2 u_{\mathbf{g}}^2}} \right). \qquad (11)$$

Theorem 4.3 follows directly from the variance expression $\sigma_j$ provided in Property 4.2; detailed steps are included in Appendix B. Using this result, we now show how to estimate the required value of $\sqrt{nk}$ to reliably identify most of the desired neurons.

We argue that for any $\xi$, if we select the top-$\xi\%$ important neurons according to the $\hat{\boldsymbol{\eta}}$ values (for some $\xi \in [0, 100]$), most of top-$\xi\%$ neurons according to the *true* $\boldsymbol{\eta}$ values will be correctly selected with high probability, when $nk$ is large enough. To formalize this, let $i_\xi = \frac{\xi}{100} d_{\boldsymbol{\eta}}$ where $d_{\boldsymbol{\eta}}$ is the number of neurons and let $\boldsymbol{\eta}_1 \geq \boldsymbol{\eta}_2 \geq \ldots \geq \boldsymbol{\eta}_{d_{\boldsymbol{\eta}}}$. Let $X_i$ be the random variable that is 1 if $\hat{\boldsymbol{\eta}}_i$ is among the top $i_\xi$ of $\{\hat{\boldsymbol{\eta}}_j\}_{j \in [d_{\boldsymbol{\eta}}]}$ and 0 otherwise. Define $p_i$ to be $p(X_i = 1)$.

We can now bound $p(X_i = 0)$ as follows. If $X_i = 0$ for $i = i_\xi$, then at least one of the values $\hat{\boldsymbol{\eta}}_{i_\xi+1}, \hat{\boldsymbol{\eta}}_{i_\xi+2}, \ldots, \hat{\boldsymbol{\eta}}_{d_{\boldsymbol{\eta}}}$, say $\hat{\boldsymbol{\eta}}_j$, must be $> \hat{\boldsymbol{\eta}}_i$. Thus, for all $i \leq i_\xi$

$$p(X_i = 0) \leq 1 - \prod_{j=i_\xi+1}^{d_{\boldsymbol{\eta}}} p(\hat{\boldsymbol{\eta}}_i > \hat{\boldsymbol{\eta}}_j) \leq 1 - p(\hat{\boldsymbol{\eta}}_i > \hat{\boldsymbol{\eta}}_{i_\xi})^{d_{\boldsymbol{\eta}}-i_\xi} \implies p_i \geq p(\hat{\boldsymbol{\eta}}_i > \hat{\boldsymbol{\eta}}_{i_\xi})^{d_{\boldsymbol{\eta}}-i_\xi}. \qquad (12)$$

The next result bounds the expected number of the "correct" neurons (or indices) chosen, in terms of $n, k$.

**Theorem 4.4.** *Let $\epsilon_\xi \in (0, 1]$ be any desired error ratio, and let $X_i$ be as defined above. Then there exist a positive constant $c_{d_{\boldsymbol{\eta}}}$ such that the inequality $\mathbb{E}[\sum_{i=1}^{i_\xi} X_i] \geq (1 - \epsilon_\xi) \cdot i_\xi$ holds, as long as*

$$\sqrt{nk} \geq c_{d_{\boldsymbol{\eta}}} \cdot \alpha\sqrt{(d_{i_1} + d_{i_2})u_{\boldsymbol{\theta}}^2 u_{\mathbf{g}}^2}. \qquad (13)$$

---

[4]We note that this is not a formal guarantee; indeed, if $k$ is too large or the dataset is too small, there is additional variance across calibration sets that will become significant.

Theorem 4.4 can be proved by using Equation 12 and Theorem 4.3; detailed steps are given in Appendix B. Intuitively, Theorem 4.4 indicates that a larger value of $nk$ leads to more accurate estimates of importance scores. This increases the likelihood of capturing truly important neurons, which in turn may result in better fine-tuned models.

Unfortunately, the above threshold of $\sqrt{nk}$ is too large in our experiments. To balance the variance of estimates and computation resources, we set $nk = 2048$ in our experiments. The error $\epsilon_{\xi=10}$ is likely to be higher than 20% under this setting, but our experimental results show that this setting is good enough to boost the fine-tuning performance.

## 5 Experimental Setup

**Main experiments:**

For language tasks, we fine-tune LLaMA-2-7B and LLaMA-3-8B in both full-precision (float32) and NF4 (Dettmers et al., 2024) on the training splits of 8 commonsense reasoning datasets (see Appendix E), and evaluate them on their respective test splits. We then assess the models' mathematical reasoning by fine-tuning them on the GSM8K (Cobbe et al., 2021) training split and evaluating performance using the EleutherAI LM Harness (Gao et al., 2021). Finally, we evaluate instruction-following ability by fine-tuning on Alpaca-GPT (Taori et al., 2023) and scoring responses with MT-Bench (Zheng et al., 2023), using Gemini 2.5 flash (Team et al., 2023) as the judge[5].

We incorporate the PEFT approaches including our SPruFT, QSPruFT, LoRA (Hu et al., 2022), QLoRA (Dettmers et al., 2024), VeRA (Kopiczko et al., 2024), DoRA (Liu et al., 2024), QDoRA, RoSA (Nikdan et al., 2024), LoftQ (Li et al., 2024), and QPiSSA (Meng et al., 2024). RoSA is chosen as the representative SpFT method and is the only SpFT due to the high memory demands of other SpFT approaches, while full fine-tuning is excluded for the same reason. We freeze Llama's classification layers and fine-tune only the linear layers in attention and FFN blocks.

For training configurations setting, we use a learning rate of $10^{-4}$ with linear decay (rate 0.01) for all methods, and we set $\alpha = 16$ and dropout rate equal to 0.1 for LoRA and its variants. All methods apply linear decay after a 3% warmup. For commonsense reasoning, we train on 2048 samples (256 per dataset) and evaluate on 500 test examples per dataset. For GSM8K, we fine-tune on 2048 random training samples and evaluate on the full test set. For Alpaca-GPT, we train the models on 1024 samples and evaluate the models on the whole MT-bench. Instruction-style prompts are used for commonsense datasets,[6] while GSM8K uses question-answering prompts. We fine-tune all models for 3 epochs.

Our framework is built on torch-pruning (Fang et al., 2023), PyTorch (Paszke et al., 2019), and HuggingFace Transformers (Wolf et al., 2020). Most experiments are conducted on a single A100-80GB GPU, except DoRA and RoSA (at 2048 max tokens) which run on an H100-96GB. We use the Adam optimizer (Kingma & Ba, 2015) and train with a fixed epoch budget without early stopping.

**Experiments for importance metrics:** In the comparison of different importance metrics, we also use our approach to fine-tune DeiT (Touvron et al., 2021), ViT (Dosovitskiy, 2020), ResNet101 (He et al., 2016), and ResNeXt101 (Xie et al., 2017) on Tiny-ImageNet (Tavanaei, 2020), CIFAR100 (Krizhevsky et al., 2009), and Caltech101 (Li et al., 2022). For these tasks, we set the fine-tuning ratio at 5%, meaning the trainable parameters are a total of 5% of the backbone plus classification layers. Following this, we discuss the computational resource requirements for fine-tuning.

For image models, the learning rate is set to $10^{-4}$ with cosine annealing decay (Loshchilov & Hutter, 2017), where the minimum learning rate is $10^{-9}$. All image models used in this study have been pre-trained on ImageNet. Note that memory efficiency is not emphasized for small-scale models, as dataset-related memory—particularly with large batch sizes—dominates consumption in these cases. The main advantage of our method in these cases is the reduced FLOPs due to fewer trainable parameters.

---

[5]We use Gemini 2.5 instead of GPT-4 due to lower cost and competitive performance.
[6]LLaMA-3 performs well with question-answering prompts, and fine-tuning yields limited gains, suggesting possible pre-training on these datasets with question-answering prompts.

Table 2: Main results of fine-tuning full precision Llama2 and Llama3. "mem" represents the memory cost in training excluding the model itself, with further details provided in Appendix D.3. #param is the number of trainable parameters. HS, OBQA, and WG represent HellaSwag, OpenBookQA, and WinoGrande datasets. All reported results for commonsense reasoning tasks are accuracies, while the results for GSM8k are the exact match score. The ablation study for different $r$ can be found in Appendix D.6. All reported results are accuracies on the corresponding tasks. **Bold** indicates the best result on the same task.

| Model, ft setting | mem | #param | BoolQ | PIQA | SIQA | HS | WG | ARC-c | ARC-e | OBQA | Avg | GSM8k |
|---|---|---|---|---|---|---|---|---|---|---|---|---|
| Llama2(7B), pretrained | - | - | 58.00 | 40.00 | 29.00 | 15.40 | 4.80 | 14.00 | 16.20 | 25.40 | 25.35 | 0.00 |
| LoRA, $r = 64$ | 23.46GB | 159.9M(2.37%) | 77.00 | **76.20** | 67.80 | 84.20 | 62.60 | 70.00 | 82.00 | **74.00** | 74.23 | 18.42 |
| VeRA, $r = 64$ | 22.97GB | 1.374M(0.02%) | 47.80 | 51.80 | 41.80 | 37.60 | 50.40 | 36.80 | 43.20 | 32.80 | 41.53 | 0.00 |
| DoRA, $r = 64$ | 44.85GB | 161.3M(2.39%) | 75.20 | 75.40 | 64.60 | 78.60 | 63.00 | 65.20 | 82.20 | 70.60 | 71.85 | 21.46 |
| RoSA, $r = 32, d = 1.2\%$ | 44.69GB | 157.7M(2.34%) | 79.80 | 73.40 | **70.20** | 76.00 | 57.00 | 68.80 | 80.80 | 71.60 | 72.20 | 21.99 |
| SPruFT, $r = 128$ | **17.62GB** | 145.8M(2.16%) | **80.00** | 75.20 | 67.60 | **85.00** | **63.40** | 70.80 | **82.40** | 71.80 | **74.53** | **22.90** |
| Llama3(8B), pretrained | - | - | 58.80 | 41.60 | 38.00 | 10.20 | 11.20 | 55.20 | 63.00 | 27.40 | 38.18 | 0.00 |
| LoRA, $r = 64$ | 30.37GB | 167.8M(2.09%) | 84.20 | 77.00 | 63.20 | 84.20 | 67.20 | 76.40 | 88.80 | 71.00 | 76.50 | 41.77 |
| VeRA, $r = 64$ | 29.49GB | 1.391M(0.02%) | 61.00 | 62.40 | 55.60 | 41.80 | 49.60 | 59.60 | 77.60 | 60.00 | 58.45 | 0.00 |
| DoRA, $r = 64$ | 51.45GB | 169.1M(2.11%) | 83.20 | **82.80** | 69.00 | **89.40** | **70.80** | 77.20 | 89.00 | 80.40 | 80.23 | 46.02 |
| RoSA, $r = 32, d = 1.2\%$ | 48.40GB | 167.6M(2.09%) | 79.00 | 81.00 | 69.20 | 84.80 | 68.60 | 79.00 | 90.40 | 78.40 | 78.80 | 45.72 |
| SPruFT, $r = 128$ | **24.49GB** | 159.4M(1.98%) | **87.60** | 77.40 | **71.40** | 85.40 | 70.20 | **79.80** | **90.80** | **81.80** | **80.55** | 46.10 |

# 6 Results and Discussion

We now present the results of fine-tuning image models and Llamas using our framework. We first apply our SPruFT to fine-tune Llama2 and Llama3 and compare the results with those obtained using LoRA and its variants. Following this, we examine the performance of our approach by utilizing various importance metrics.

Table 3: Comparing *fine-tuning all linear layers* with *fine-tuning only the FFN* on commonsense reasoning tasks. [‡] indicates that we freeze the layers for queue, key, and value projection. Full table please refer to Table 19 in Appendix D.7.

| Models | | Llama2(7B) | | | Llama3(8B) | |
|---|---|---|---|---|---|---|
| Setting | mem | #param | Common(Avg) | mem | #param | Common(Avg) |
| LoRA, $r = 16$ | 21.64GB | 40.0M(0.59%) | 72.48 | 28.86GB | 41.9M(0.52%) | **78.95** |
| LoRA[‡], $r = 32$ | **17.95GB** | 54.8M(0.81%) | **72.90** | **25.28GB** | 65.0M(0.81%) | 78.35 |
| LoRA, $r = 32$ | 22.21GB | 80.0M(1.19%) | 72.03 | 29.37GB | 83.9M(1.04%) | **79.50** |
| LoRA[‡], $r = 64$ | **18.81GB** | 109.6M(1.63%) | **73.00** | **26.04GB** | 130.0M(1.62%) | 77.73 |
| SPruFT, $r = 32$ | 15.57GB | 36.4M(0.54%) | 72.25 | 22.62GB | 39.8M(0.50%) | 78.53 |
| SPruFT[‡], $r = 64$ | **14.67GB** | 47.7M(0.71%) | **72.45** | **21.81GB** | 54.5M(0.68%) | **79.05** |
| SPruFT, $r = 64$ | 16.20GB | 72.9M(1.08%) | 72.65 | 23.23GB | 79.7M(0.99%) | 77.88 |
| SPruFT[‡], $r = 128$ | **15.58GB** | 95.4M(1.42%) | **73.98** | **22.71GB** | 109.1M(1.36%) | **80.45** |

## 6.1 Main Results of LLM

We apply our SPruFT method to fine-tune Llama2-7B and Llama3-8B, comparing the results with those obtained through LoRA and its variants. We select the magnitude of the neuron vector as the importance metric due to its low memory requirements, simplicity, and widely tested effectiveness. In contrast, gradient-based metrics like Taylor and Hessian are as memory-intensive as full LLM fine-tuning. While Wanda (Sun et al., 2024) and AWQ (Lin et al., 2024) offer two activation-based metrics to evaluate neuron importance, for pruning LLMs, they require one epoch of data forwarding and significantly more memory than inference to compute the activation-based importance[7].

---

[7]We encountered an OOM error when using Wanda's official implementation. When pruning a neural network, each layer computes activation vector's norm and is pruned immediately, gradually reducing the model size. However, in the fine-tuning process, the model size remains unchanged. Additionally, storing the activation values for computing importance scores further increases memory consumption, making memory cost significantly higher than when using activation for pruning.

Table 2 demonstrates the exceptional memory efficiency of our approach[8] while achieving comparable accuracy. As shown, the accuracies of fine-tuned models remain similar across most PEFT methods, while memory usage varies significantly. VeRA, despite having significantly fewer trainable parameters, shows noticeably lower accuracy. Notably, our approach consistently requires substantially less memory than all other PEFT methods listed in the table.

Table 4: MT-Bench results of fine-tuning full precision Llama2 and Llama3 on Alpaca-GPT4. **Bold** indicates the best result on the same task.

| Model, ft setting | Coding | Extraction | Humanities | Math | Reasoning | Roleplay | Stem | Writing | Avg |
|---|---|---|---|---|---|---|---|---|---|
| Llama2(7B), pretrained | 1.00 | 2.63 | 1.65 | 1.50 | 1.59 | 1.65 | 1.70 | 1.55 | 1.66 |
| LoRA, $r = 64$ | 1.50 | 2.59 | **5.21** | 2.43 | 3.27 | **5.40** | **4.59** | 3.84 | 3.60 |
| VeRA, $r = 64$ | 1.47 | 2.75 | 2.30 | 1.42 | 3.06 | 2.55 | 3.06 | 2.10 | 2.34 |
| DoRA, $r = 64$ | 1.13 | 2.55 | 4.22 | 1.93 | 3.50 | 4.15 | 4.35 | **4.35** | 3.27 |
| RoSA, $r = 32, d = 1.2\%$ | 1.73 | **3.35** | 5.00 | **2.64** | **4.90** | 4.80 | 3.94 | 3.84 | **3.78** |
| SPruFT, $r = 128$ | **1.82** | 2.55 | 4.80 | 2.08 | 4.07 | 4.79 | 4.50 | 3.53 | 3.52 |
| LoRA$^{\ddagger}$, $r = 64$ | **2.36** | 1.89 | 4.89 | 1.92 | 3.36 | 4.30 | 4.05 | **3.84** | 3.33 |
| VeRA$^{\ddagger}$, $r = 64$ | 1.47 | 2.50 | 2.15 | 1.39 | 3.06 | 2.90 | 2.32 | 1.85 | 2.20 |
| DoRA$^{\ddagger}$, $r = 64$ | 1.10 | 1.75 | 4.25 | 2.46 | 3.62 | **4.89** | 4.21 | 3.58 | 3.23 |
| RoSA$^{\ddagger}$, $r = 32, d = 1.2\%$ | 2.00 | **3.11** | **5.60** | **2.50** | **4.43** | 4.50 | 4.11 | 3.74 | **3.75** |
| SPruFT$^{\ddagger}$, $r = 128$ | 1.08 | 2.10 | 4.60 | 1.15 | 3.80 | 4.22 | **4.33** | 3.42 | 3.09 |
| Llama3(8B), pretrained | 3.56 | 5.56 | 2.95 | 1.32 | 2.00 | 2.84 | 3.56 | 3.75 | 3.19 |
| LoRA, $r = 64$ | 2.88 | 3.88 | 5.05 | 4.33 | 4.25 | 5.89 | 5.72 | 5.22 | 4.65 |
| VeRA, $r = 64$ | 3.18 | 5.28 | 4.05 | 1.00 | 2.39 | 3.55 | 4.72 | 3.00 | 3.40 |
| DoRA, $r = 64$ | 3.82 | 4.85 | **6.40** | **5.00** | 4.00 | **6.21** | 5.88 | 4.65 | 5.10 |
| RoSA, $r = 32, d = 1.2\%$ | 3.50 | **7.31** | 6.30 | 4.31 | **4.58** | 5.75 | **7.19** | 5.30 | **5.53** |
| SPruFT, $r = 128$ | **3.88** | 5.11 | 6.11 | 3.83 | 4.21 | 5.35 | 6.44 | **5.40** | 5.04 |
| LoRA$^{\ddagger}$, $r = 64$ | 3.75 | 4.78 | 6.15 | 4.42 | **4.75** | 5.15 | 6.06 | **5.95** | 5.12 |
| VeRA$^{\ddagger}$, $r = 64$ | 3.33 | 5.32 | 4.35 | 1.00 | 1.76 | 3.50 | 4.25 | 4.00 | 3.44 |
| DoRA$^{\ddagger}$, $r = 64$ | 3.90 | 5.56 | **6.75** | 4.42 | 3.82 | **6.50** | **6.88** | 5.05 | **5.36** |
| RoSA$^{\ddagger}$, $r = 32, d = 1.2\%$ | 3.63 | **5.88** | 6.05 | **4.43** | 2.92 | 5.32 | 6.87 | 5.30 | 5.05 |
| SPruFT$^{\ddagger}$, $r = 128$ | **4.75** | 4.78 | 5.16 | 3.67 | 3.31 | 6.25 | 6.06 | 5.00 | 4.87 |

We then demonstrate that strategically assigning trainable parameters saves more memory than merely reducing them, without compromising accuracy on commonsense reasoning tasks. Section 3 highlights the importance of the computation graph in memory consumption. Strategically assigning trainable parameters can be an effective solution. For instance, Shi et al. (2024) suggest that layers such as output, down, up, and gate projections contribute more significantly than queue, key, and value projections, which leads to the strategy of freezing queue, key, and value projections. We compare *fine-tuning all linear layers* and *fine-tuning only the FFN and output-project layer*, with results shown in Table 3. The former requires more memory for the same number of trainable parameters, as distributing trainable parameters across the model increases the need for caching intermediate values. Table 3 shows an exceptional memory saving by freezing some layers in attention blocks. In addition, the accuracy remains nearly unchanged. Given that Llama models have been pre-trained on extensive datasets, their attention layers likely already capture crucial patterns for token interactions.

We further evaluate models' instruction-following abilities using MT-Bench. As shown in Table 4, SPruFT with magnitude-based importance falls short of other methods. However, incorporating ZOTaylor significantly improves performance: Table 5 (Section 6.2) shows that SPruFT with ZOTaylor outperforms most approaches on instruction-following tasks, ranking second in all cases. Only RoSA performs better overall, but it requires substantially more memory than our method.

## 6.2 Importance Metrics

We apply various importance metrics to fine-tune Llamas and image models using our approach and report the results to compare their performance. As shown in Table 5 and Table 6, Quantile-Mean Taylor and ZOTaylor offer slight improvements over other importance metrics. For image tasks, while the differences among importance metrics are not substantial, the results consistently indicate that Quantile-Mean Taylor slightly outperforms standard Taylor importance. Additionally, both Quantile-Mean Taylor and standard Taylor importance outperform magnitude-based importance.

---

[8]Also refer to Table 16 in Appendix D.6, even with $r = 128$, our method's memory usage remains significantly lower than that of LoRA with $r = 8$.

Similarly, in the cases of Llama2 and Llama3, our findings suggest that ZOTaylor provides a slight performance boost for fine-tuned models. This improvement is likely due to ZOTaylor's ability to capture richer data information, whereas magnitude-based importance tends to focus more on identifying generally important neurons. However, the observed performance gain remains modest, potentially due to the variance of the estimates, as discussed in Section 4.3.2. Beyond these observations, an interesting finding is that models fine-tuned with random row selection significantly outperform VeRA, likely suggesting that overly aggressive parameter reduction can substantially degrade performance.

Table 5: Importance evaluation (random (rd) vs $\ell^2$ vs ZOTaylor (ZO)) for Llama2 and Llama3 on commonsense reasoning tasks, GSM8k, and MT-Bench. We also present the results of freezing queue-, key-, and value-projection in this table ($\ddagger$). Full table with different ranks for commonsense tasks please refer to Table 17 and Table 18 in Appendix D.6.

| Model | Llama2(7B) | | | Llama3(8B) | | |
|---|---|---|---|---|---|---|
| Settings | Common(Avg) | GSM8K | MT-Bench(Avg) | Common(Avg) | GSM8k | MT-Bench(Avg) |
| $r = 128$, rd | 69.80 | 22.52 | 3.45 | 76.08 | 42.53 | 4.40 |
| $r = 128$, $\ell^2$ | **74.53** | 22.90 | 3.52 | 80.55 | 46.10 | 5.04 |
| $r = 128$, ZO | 74.35 | **23.50** | **3.62** | **81.28** | **55.12** | **5.34** |
| $r = 128^{\ddagger}$, rd | 72.08 | 21.08 | 3.60 | 77.20 | 39.50 | 5.05 |
| $r = 128^{\ddagger}$, $\ell^2$ | 73.98 | 19.48 | 3.09 | 80.45 | 42.53 | 4.87 |
| $r = 128^{\ddagger}$, ZO | **74.30** | **24.34** | **3.66** | **81.80** | **46.02** | **5.21** |

Table 6: Importance metrics on fine-tuning image models by our SPruFT for 5 epochs. FFT, $\ell^2$, Taylor, and QMTaylor represent full fine-tuning, the magnitude, Taylor importance, and Quantiles-Mean Taylor importance (Equation 8). Note that QMTaylor is not applied to fine-tuning Caltech101 due to its significantly imbalanced labels. All reported results are validation accuracies. **Bold** indicates the superior results achieved through different importance metrics.

| model | imp | CIFAR100 | Tiny-ImageNet | Caltech101 |
|---|---|---|---|---|
| DeiT | FFT | 90.18 | 87.55 | 97.33 |
| | $\ell^2$ | 88.05 | 89.31 | 95.01 |
| | Taylor | 88.70 | 89.69 | **95.41** |
| | Hessian | 88.73 | 89.66 | 95.10 |
| | QMTaylor | **89.37** | **89.75** | - |
| ViT | FFT | 90.13 | 88.45 | 97.16 |
| | $\ell^2$ | 87.13 | 90.78 | 92.69 |
| | Taylor | 88.06 | 90.87 | **93.96** |
| | Hessian | 87.63 | 90.56 | 93.70 |
| | QMTaylor | **88.51** | **90.90** | - |
| RN | FFT | 86.21 | 77.78 | 96.50 |
| | $\ell^2$ | 82.25 | 79.83 | **93.13** |
| | Taylor | 82.36 | 79.66 | 92.56 |
| | Hessian | 82.50 | 79.67 | 92.74 |
| | QMTaylor | **83.50** | **80.02** | - |
| RNX | FFT | 87.30 | 79.51 | 97.07 |
| | $\ell^2$ | 86.12 | 83.88 | 95.71 |
| | Taylor | 85.94 | 83.88 | **95.84** |
| | Hessian | 85.77 | **84.53** | 95.63 |
| | QMTaylor | **86.93** | 84.17 | - |

## 6.3 Dropout Comparison and $r$ Sensitivity Analysis

We investigate the role of dropout in SpFT and LoRA to assess whether comparing our approach without dropout to LoRA with dropout is appropriate. While prior works on SpFT (Sung et al., 2021; Guo et al., 2021; Ansell et al., 2022; Nikdan et al., 2024; Guo et al., 2024b; Panda et al., 2024) have already compared SpFT without dropout to LoRA variants with dropout and found that SpFT is generally less prone to overfitting, we perform additional experiments to ensure that this comparison is fair. The key distinction is that SpFT itself constitutes a form of sparse training, which inherently provides a regularization effect. Introducing dropout into SpFT may therefore lead to over-regularization. By contrast, LoRA and its variants often benefit from dropout, as it can help mitigate overfitting. The results in Table 7 and Table 8 confirm

Table 7: The influence of dropout on LoRA. Details on memory footprint and #param are provided in Table 16.

| LoRA | $r=8$, 0.25% | | $r=16$, 0.5% | | $r=32$, 1% | | $r=64$, 2% | | $r=128$, 4% | | $r=256$, 8% | |
|---|---|---|---|---|---|---|---|---|---|---|---|---|
| | common | gsm8k | common | gsm8k | common | gsm8k | common | gsm8k | common | gsm8k | common | gsm8k |
| Llama2(7B) | | | | | | | | | | | | |
| w/ dropout | **73.55** | **15.39** | 72.48 | **19.86** | 72.03 | **19.56** | **74.23** | 18.42 | **74.53** | **22.90** | **73.90** | **18.20** |
| w/o dropout | 72.63 | 15.16 | **72.98** | 19.64 | **72.70** | 17.82 | 73.15 | **18.88** | 72.78 | 21.30 | 73.65 | 17.36 |
| Llama3(8B) | | | | | | | | | | | | |
| w/ dropout | **80.88** | **35.25** | 78.95 | **42.15** | **79.50** | **45.56** | 76.50 | 41.77 | **80.18** | **42.23** | **77.73** | **38.82** |
| w/o dropout | 77.80 | 35.10 | **80.80** | 38.82 | 78.23 | 40.79 | **79.63** | **44.05** | 78.30 | 40.56 | 75.73 | 35.71 |

Table 8: The influence of dropout on our approach, including importance metrics random (rd), $\ell^2$, and ZOTaylor (ZO). Details on memory footprint and #param are provided in Table 16.

| Ours | $r=16$, 0.25% | | $r=32$, 0.5% | | $r=64$, 1% | | $r=128$, 2% | | $r=256$, 4% | | $r=512$, 8% | |
|---|---|---|---|---|---|---|---|---|---|---|---|---|
| | common | gsm8k | common | gsm8k | common | gsm8k | common | gsm8k | common | gsm8k | common | gsm8k |
| Llama2(7B) | | | | | | | | | | | | |
| rd, w/ dropout | 66.70 | 14.71 | 69.53 | 16.45 | 68.75 | 19.94 | 68.03 | 21.61 | 69.20 | **20.32** | 65.00 | 14.48 |
| rd, w/o dropout | **66.75** | **15.85** | **69.80** | **18.35** | **69.45** | **20.39** | **69.80** | **22.52** | **69.23** | 18.88 | **68.60** | **17.06** |
| $\ell^2$, w/ dropout | 67.00 | 15.09 | 70.50 | 17.66 | 70.60 | **21.53** | 72.83 | 22.59 | 73.20 | 19.03 | 73.60 | 17.59 |
| $\ell^2$, w/o dropout | **67.83** | **15.31** | **72.25** | **18.27** | **72.65** | 21.00 | **74.53** | **22.90** | **73.68** | **20.02** | **73.93** | **17.74** |
| ZO, w/ dropout | 68.38 | 15.77 | 71.80 | 23.43 | 70.28 | 21.00 | 73.83 | 22.59 | 74.63 | 22.21 | 73.23 | **17.13** |
| ZO, w/o dropout | **70.70** | **17.82** | **72.35** | **23.81** | **72.80** | **22.06** | **74.35** | **23.50** | 73.95 | **24.94** | **73.45** | 16.07 |
| Llama3(8B) | | | | | | | | | | | | |
| rd, w/ dropout | 76.50 | 32.15 | 76.35 | **40.03** | 73.78 | 46.85 | **76.55** | **43.29** | 74.80 | 37.45 | 74.10 | 33.13 |
| rd, w/o dropout | **78.00** | **35.03** | **77.75** | 36.54 | **75.18** | **48.22** | 76.08 | 42.53 | **76.95** | **38.29** | **75.68** | **36.62** |
| $\ell^2$, w/ dropout | 73.53 | 31.46 | 77.75 | **43.29** | 77.03 | 43.75 | 79.78 | 45.79 | 77.70 | 36.77 | 77.93 | 34.72 |
| $\ell^2$, w/o dropout | **75.43** | **33.59** | **78.53** | 41.70 | **77.88** | **46.25** | **80.55** | **46.10** | **78.80** | **38.89** | **78.70** | **37.45** |
| ZO, w/ dropout | 78.10 | 36.39 | 77.05 | **47.38** | **78.00** | 49.73 | 79.73 | 52.08 | 79.25 | 49.96 | 79.18 | 42.15 |
| ZO, w/o dropout | **78.33** | **37.53** | **78.83** | 44.50 | **78.00** | **50.19** | **81.28** | **55.12** | **80.20** | **50.34** | **79.98** | **47.54** |

these tendencies: SpFT generally does not rely on dropout, whereas LoRA shows some improvements when dropout is applied. In fact, dropout brings little to no benefit in our method and can even slightly degrade performance, while in LoRA it typically yields gains.

Table 7 and Table 8 also illustrate the effect of sparsity ratio and rank. When the sparsity ratio is very small (0.25%), our approach performs less effectively, and LoRA achieves slightly better results. As the sparsity ratio increases, performance improvements diminish, suggesting that the range $[0.5\%, 2\%]$ may be the most suitable. Somewhat surprisingly, performance decreases when the rank $r$ is large (256 for LoRA and 512 for our approach). We attribute this to the relatively small size of the fine-tuning dataset compared with the model size and the pre-training corpus. With more trainable parameters, the model is more likely to overfit. This observation is also consistent with the fact that many prior works using LoRA select rank values between 32 and 128, further supporting the view that PEFT can sometimes outperform full fine-tuning.

Table 9: Main results of fine-tuning NF4-quantized LLaMA-2 and LLaMA-3. QSPruFT$^*$ denotes our approach with $\Delta\mathbf{W}$ initialized randomly, while QSPruFT$^\dagger$ uses $\Delta\mathbf{W}$ initialized with the corresponding rows from the quantization residual $\mathbf{W}^{res}$.

| Model, ft setting | BoolQ | PIQA | SIQA | HS | WG | ARC-c | ARC-e | OBQA | Avg | GSM8k |
|---|---|---|---|---|---|---|---|---|---|---|
| Llama2(7B), pretrained | 37.00 | 37.00 | 19.20 | 20.00 | 4.40 | 15.40 | 17.80 | 20.80 | 21.48 | 0.00 |
| pretrained-NF4 | 58.00 | 40.00 | 29.00 | 15.40 | 4.80 | 14.00 | 16.20 | 25.40 | 25.35 | 0.00 |
| QLoRA, $r=64$ | 75.20 | 74.80 | 66.80 | 73.80 | 63.60 | 63.40 | 77.60 | 65.40 | 70.08 | 18.88 |
| QDoRA, $r=64$ | 77.40 | 72.00 | 68.40 | 81.60 | 64.60 | 64.60 | 80.20 | 70.60 | 72.25 | **25.63** |
| LoftQ, $r=64$ | 73.80 | **73.20** | **72.60** | 81.60 | 64.20 | **67.60** | 82.00 | 71.60 | 73.33 | 20.55 |
| QPiSSA, $r=64$ | 79.40 | 72.40 | 68.60 | 77.40 | 65.00 | 67.40 | 78.60 | 69.80 | 72.33 | 21.08 |
| QSPruFT$^*$, $r=128$ | 78.00 | 72.60 | 69.00 | 79.40 | 63.20 | 67.20 | 79.60 | **74.80** | 72.98 | 20.32 |
| QSPruFT$^\dagger$, $r=128$ | **82.00** | 71.80 | 69.20 | **83.20** | **65.40** | 67.20 | 79.20 | 72.40 | **73.80** | 22.14 |
| Llama3(8B), pretrained | 58.80 | 41.60 | 38.00 | 10.20 | 11.20 | 55.20 | 63.00 | 27.40 | 38.18 | 0.00 |
| pretrained-NF4 | 52.00 | 37.80 | 11.60 | 8.40 | 0.20 | 48.80 | 53.80 | 16.20 | 28.60 | 0.00 |
| QLoRA, $r=64$ | 77.40 | 81.60 | **72.80** | 87.60 | 69.60 | 75.80 | 90.20 | 78.60 | 79.20 | 41.17 |
| QDoRA, $r=64$ | 79.00 | 79.20 | 68.20 | 82.20 | 66.40 | 75.80 | 87.40 | 77.20 | 76.93 | 46.17 |
| LoftQ, $r=64$ | 87.00 | **82.80** | 69.40 | **90.20** | 62.40 | 74.80 | 91.00 | **79.60** | 79.65 | 44.96 |
| QPiSSA, $r=64$ | 87.60 | 82.00 | 71.00 | 82.60 | **71.80** | **77.20** | 89.60 | 77.20 | 79.88 | 45.87 |
| QSPruFT$^*$, $r=128$ | 89.80 | 79.40 | 65.40 | 89.80 | 68.40 | 76.20 | **91.80** | 78.00 | 79.85 | 47.76 |
| QSPruFT$^\dagger$, $r=128$ | **90.40** | 82.40 | 68.60 | 89.40 | 69.20 | 76.80 | 90.80 | 76.40 | **80.50** | **49.13** |

### 6.4 Results of Quantized Models

Tables 9 and 10 present the results of fine-tuning NF4-quantized models using various PEFT methods[9]. In these quantized experiments, we use magnitude-based importance metric to save processing time, as quantized parameters increase latency during training and inference. The results show that our method, with $\Delta\mathbf{W}$ initialized from $\mathbf{W}^{res}$, consistently achieves strong performance across tasks. These findings support that SPruFT is a simple, effective, and memory-efficient fine-tuning strategy.

Table 10: Main results of fine-tuning NF4-quantized LLaMA-2 and LLaMA-3. QSPruFT$^*$ denotes our approach with $\Delta\mathbf{W}$ initialized randomly, while QSPruFT$^\dagger$ uses $\Delta\mathbf{W}$ initialized with the corresponding rows from the quantization residual $\mathbf{W}^{res}$.

| Model, ft setting | Coding | Extraction | Humanities | Math | Reasoning | Roleplay | Stem | Writing | Avg |
|---|---|---|---|---|---|---|---|---|---|
| Llama2(7B), pretrained | 1.00 | 2.63 | 1.65 | 1.50 | 1.59 | 1.65 | 1.70 | 1.55 | 1.66 |
| pretrained-NF4 | 1.00 | 2.84 | 1.60 | 1.19 | 1.67 | 1.85 | 1.80 | 1.40 | 1.67 |
| QLoRA, $r = 64$ | 1.00 | 3.05 | 4.74 | 1.79 | 3.23 | 5.15 | 4.69 | 3.55 | 3.40 |
| QDoRA, $r = 64$ | 1.46 | 3.78 | 5.42 | 1.92 | 3.63 | 4.63 | 4.63 | 4.05 | 3.69 |
| LoftQ, $r = 64$ | 1.46 | 3.90 | 5.55 | **2.45** | **4.69** | 4.60 | 4.37 | 3.89 | 3.87 |
| QSPruFT$^*$, $r = 128$ | 0.88 | 3.75 | **5.65** | 2.29 | 2.77 | 4.58 | 3.82 | 4.40 | 3.52 |
| QSPruFT$^\dagger$, $r = 128$ | **1.92** | **4.70** | 5.45 | 1.93 | 4.57 | **5.20** | **4.78** | **4.60** | **4.14** |
| Llama3(8B), pretrained | 3.56 | 5.56 | 2.95 | 1.32 | 2.00 | 2.84 | 3.56 | 3.75 | 3.19 |
| pretrained-NF4 | 1.57 | 4.32 | 2.35 | 1.75 | 2.50 | 2.80 | 3.11 | 2.21 | 2.58 |
| QLoRA, $r = 64$ | 4.43 | 5.41 | 4.74 | 3.00 | **4.79** | 5.65 | 4.74 | 4.68 | 4.68 |
| QDoRA, $r = 64$ | 3.56 | 6.26 | 6.00 | 4.43 | 3.83 | **5.95** | 5.28 | 5.06 | 5.05 |
| LoftQ, $r = 64$ | 4.30 | **7.00** | 6.60 | **5.55** | 3.73 | 5.90 | 5.60 | **6.30** | **5.62** |
| QSPruFT$^*$, $r = 128$ | 2.43 | 3.74 | 4.26 | 2.90 | 3.40 | 3.53 | 3.89 | 3.95 | 3.51 |
| QSPruFT$^\dagger$, $r = 128$ | **4.50** | 5.41 | **7.06** | 4.00 | 4.62 | 5.42 | **5.79** | 5.80 | 5.32 |

## 7 Conclusions and Future Work

We propose a parameter-efficient fine-tuning (PEFT) framework that integrates various techniques and importance metrics from model compression research to enhance sparse fine-tuning (SpFT). Using our method, we can fine-tune LLMs and vision transformers using significantly less computation resources than the popular LoRA (Low-Rank Adaptation) technique, while achieving similar accuracy. We also explore the effects of using different importance metrics. There are several future directions: (1) For importance metrics, while Quantile-Mean Taylor shows slight improvements, these gains are relatively minor compared to the standard Taylor metric in some cases of DeiT and ViT. We may wish to explore better metrics for classification tasks with a large number of labels. (2) Developing memory-efficient importance metrics for LLMs is another future direction. While Zeroth-Order Taylor is effective for incorporating data-specific information without requiring large memory, the large variance of estimate is a challenge. Although we reduce the variance effectively by increasing the number of estimations, exploring a simple method to reduce variance without increasing estimation times is essential for further advancements in this field. (3) Our results show that fine-tuning a small number of neurons can significantly improve model performance on downstream tasks. This observation naturally raises the question: do the selected neurons play a distinctive role in specific tasks? This question is related to the explainability of neural networks, which is an extensive area of research. It will be interesting to understand if (and how) individual neurons chosen for fine-tuning contribute to the new task.

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

## A  Importance Metrics

**Taylor importance** is the Taylor expansion of the difference between the loss of the model with and without the target neuron: (we denote $\boldsymbol{\theta}^{(i)}$ by $\mathbf{w}$ here for convenience.)

$$
\begin{aligned}
\boldsymbol{\eta}_i &= L(\mathcal{D}, \acute{F}_i) - L(\mathcal{D}, F) \\
&\approx -\mathbf{w}^\top \nabla_{\mathbf{w}} L(\mathcal{D}, F) + \frac{1}{2}\mathbf{w}^\top \nabla_{\mathbf{w}}^2 L(\mathcal{D}, F)\mathbf{w} + \mathcal{O}(\nabla_{\mathbf{w}}^3 L(\mathcal{D}, F)) \\
&\overset{(*)}{\approx} \frac{1}{2}\mathbf{w}^\top \nabla_{\mathbf{w}}^2 L(\mathcal{D}, F)\mathbf{w} + \mathcal{O}(\nabla_{\mathbf{w}}^3 L(\mathcal{D}, F)) \\
&\overset{(**)}{\approx} \frac{1}{2}(G\mathbf{w})^\top (G\mathbf{w}) + \mathcal{O}(\nabla_{\mathbf{w}}^3 L(\mathcal{D}, F)),
\end{aligned}
$$

where $G = \nabla_{\mathbf{w}} L(\mathcal{D}, F)$. (**) is from the result of Fisher information Rissanen (1996):

$$
\nabla_{\mathbf{w}}^2 L(\mathcal{D}, F) \approx \nabla_{\mathbf{w}} L(\mathcal{D}, F)^\top \nabla_{\mathbf{w}} L(\mathcal{D}, F).
$$

Note that (*) is from $\nabla_{\mathbf{w}} L(\mathcal{D}, F) \approx 0$, as removing one channel/neuron from a large neural network typically results in only a negligible reduction in loss. To efficiently compute $\boldsymbol{\eta}_i$, the equation can be further derived as:

$$
\boldsymbol{\eta}_i \approx (G\mathbf{w})^\top (G\mathbf{w}) = \sum_j \left( \frac{1}{|\mathcal{D}|} \sum_{\mathbf{x}\in\mathcal{D}} \frac{\partial L(\mathbf{x}, F)}{\partial \mathbf{w}_j} \mathbf{w}_j \right)^2 \tag{14}
$$

$$
\approx \sum_j \left| \frac{1}{|\mathcal{D}|} \sum_{\mathbf{x}\in\mathcal{D}} \frac{\partial L(\mathbf{x}, F)}{\partial \mathbf{w}_j} \mathbf{w}_j \right|^2, \tag{15}
$$

where $\mathbf{w} = (\mathbf{w}_1, \ldots, \mathbf{w}_j, \ldots)$. Thus, people often compute $\sum_j \left| \frac{1}{|\mathcal{D}|} \sum_{\mathbf{x}\in\mathcal{D}} \frac{\partial L(\mathbf{x}, F)}{\partial \mathbf{w}_j} \mathbf{w}_j \right|$ without taking square to evaluate the importance score. In this study, we describe this as Taylor importance and consider eq 14 as hessian importance.

**Magnitude importance** is the $\ell_2$-norm of the neuron vector computed as $\sqrt{\sum_j \mathbf{w}_j^2}$.

## B   Missing Proofs

**Proof of Property 4.2**

*Proof.* To prove Property 4.2, we first calculate the expectation and variance of SPSA. For convenience, we denote $[\nabla L(\boldsymbol{\theta}, \mathcal{D})]$ as $\mathbf{g}$. Then, the expectation is as follows:

$$\mathbb{E}[\hat{\mathbf{g}}] \approx \mathbb{E}[\mathbf{z}\mathbf{z}^\top \nabla L(\boldsymbol{\theta}, \mathcal{D})] = \mathbb{E}[\mathbf{z}\mathbf{z}^\top]\nabla L(\boldsymbol{\theta}, \mathcal{D}) = \mathbf{I}_d \nabla L(\boldsymbol{\theta}, \mathcal{D}).$$

The variance can then be derived as follows:

$$\mathrm{Var}[\hat{\mathbf{g}}_i] \approx \mathrm{Var}[\mathbf{z}_i(\mathbf{z}^\top\mathbf{g})] = \mathbb{E}[(\mathbf{z}_i(\mathbf{z}^\top\mathbf{g}))^2] - \mathbb{E}[\mathbf{z}_i(\mathbf{z}^\top\mathbf{g})]^2 = \mathbb{E}[\mathbf{z}_i^2(\sum_{i=1}^d \mathbf{z}_i\mathbf{g}_i)^2] - \mathbb{E}[\mathbf{z}_i(\sum_{l=1}^d \mathbf{z}_l\mathbf{g}_l)]^2$$

$$= \mathbb{E}[\mathbf{z}_i^2(\sum_{i=1}^d \mathbf{z}_i\mathbf{g}_i)^2] - \mathbb{E}[\mathbf{z}_i^2\mathbf{g}_i + (\mathbf{z}_i \sum_{l\neq i, l\in[d]} \mathbf{z}_l\mathbf{g}_l)]^2$$

$$= \mathbb{E}[\mathbf{z}_i^2(\sum_{i=1}^d \mathbf{z}_i\mathbf{g}_i)^2] - \left(\underbrace{\mathbb{E}[\mathbf{z}_i^2]}_{1}\mathbf{g}_i + \sum_{l\neq i, l\in[d]} \underbrace{\mathbb{E}[\mathbf{z}_i\mathbf{z}_l]}_{0}\mathbf{g}_l\right)^2 = \mathbb{E}[\mathbf{z}_i^2(\sum_{i=1}^d \mathbf{z}_i\mathbf{g}_i)^2] - \mathbf{g}_i^2$$

$$= \mathbb{E}[\mathbf{z}_i^2(\mathbf{z}_i\mathbf{g}_i + \sum_{l\neq i, l\in[d]} \mathbf{z}_l\mathbf{g}_l)^2] - \mathbf{g}_i^2$$

$$= \mathbb{E}[\mathbf{z}_i^2(\mathbf{z}_i^2\mathbf{g}_i^2 + 2\mathbf{z}_i\mathbf{g}_i(\sum_{l\neq i, l\in[d]} \mathbf{z}_l\mathbf{g}_l) + (\sum_{l\neq i, l\in[d]} \mathbf{z}_l\mathbf{g}_l)^2] - \mathbf{g}_i^2$$

$$= \mathbb{E}[\mathbf{z}_i^4\mathbf{g}_i^2 + 2\mathbf{z}_i^3\mathbf{g}_i(\sum_{l\neq i, l\in[d]} \mathbf{z}_l\mathbf{g}_l) + \mathbf{z}_i^2(\sum_{l\neq i, l\in[d]} \mathbf{z}_l\mathbf{g}_l)^2] - \mathbf{g}_i^2$$

$$= \mathbb{E}[\mathbf{z}_i^4\mathbf{g}_i^2] + 2\mathbb{E}[\mathbf{z}_i^3\mathbf{g}_i]\mathbb{E}[\sum_{l\neq i, l\in[d]} \mathbf{z}_l\mathbf{g}_l] + \mathbb{E}[\mathbf{z}_i^2]\mathbb{E}[(\sum_{l\neq i, l\in[d]} \mathbf{z}_l\mathbf{g}_l)^2] - \mathbf{g}_i^2$$

$$= \mathbf{g}_i^2\underbrace{\mathbb{E}[\mathbf{z}_i^4]}_{3} + 2\mathbf{g}_i\mathbb{E}[\mathbf{z}_i^3]\sum_{l\neq i, l\in[d]} \mathbf{g}_l\underbrace{\mathbb{E}[\mathbf{z}_l]}_{0} + \underbrace{\mathbb{E}[\mathbf{z}_i^2]}_{1}(\sum_{l\neq i, l\in[d]} \mathbf{g}_l^2\underbrace{\mathbb{E}[\mathbf{z}_l^2]}_{1} + \sum_{k\neq l\neq i, k,l\in[d]} \mathbf{g}_k\mathbf{g}_l\underbrace{\mathbb{E}[\mathbf{z}_k\mathbf{z}_l]}_{0}) - \mathbf{g}_i^2$$

$$= 3\mathbf{g}_i^2 + (\sum_{l\neq i, l\in[d]} \mathbf{g}_l^2) - \mathbf{g}_i^2 = \mathbf{g}_i^2 + \sum_{l=1}^d \mathbf{g}_l^2$$

$\mathbb{E}[\mathbf{z}_l] = 0$, $\mathbb{E}[\mathbf{z}_i^2] = \mathrm{Var}[\mathbf{z}_i] = 1$, and $\mathbb{E}[\mathbf{z}_k\mathbf{z}_l]_{l\neq k} = 0$ are because $\mathbf{z} \sim \mathcal{N}(0, \mathbf{I}_d)$. $\mathbb{E}[\mathbf{z}_i^4] = 3$ is obtained from the moment generating function of standard normal:

$$\mathbb{E}[\mathbf{z}_i^4] = \frac{d^4}{dt^4}(e^{\frac{t^2}{2}})\big|_{t=0} = (3e^{\frac{t^2}{2}} + 6t^2 e^{\frac{t^2}{2}} + t^4 e^{\frac{t^2}{2}})\big|_{t=0} = 3.$$

Thus, the expectation and variance of $n$-SPSA estimate are ($\hat{\mathbf{g}}_i^{(j)}$ is the $j^{th}$ estimate of $\hat{\mathbf{g}}_i$ here):

$$\mathbb{E}[\frac{\sum_{j=1}^n \hat{\mathbf{g}}_i^{(j)}}{n}] = \frac{\sum_{j=1}^n \mathbb{E}[\hat{\mathbf{g}}_i^{(j)}]}{n} = \frac{\sum_{j=1}^n \mathbf{g}_i}{n} = \mathbf{g}_i,$$

$$\mathrm{Var}[\frac{\sum_{j=1}^n \hat{\mathbf{g}}_i^{(j)}}{n}] = \frac{\sum_{j=1}^n \mathrm{Var}[\hat{\mathbf{g}}_i^{(j)}]}{n^2} = \frac{\mathrm{Var}[\hat{\mathbf{g}}_i]}{n} = \frac{\mathbf{g}_i^2 + \sum_{l=1}^d \mathbf{g}_l^2}{n}.$$

Then, we are going to prove the consistency of $n$-SPSA estimate. Given any small $\epsilon > 0$, we can derive the following inequality by Chebyshev's Inequality (Bienaymé, 1853; Tchébychef, 1867):

$$\mathrm{Pr}[|\frac{\sum_{j=1}^n \hat{\mathbf{g}}^{(j)}}{n} - \mathbf{g}| > \epsilon] \leq \frac{\mathrm{Var}[\frac{\sum_{j=1}^n \hat{\mathbf{g}}^{(j)}}{n}]}{\epsilon^2} = \frac{\mathbf{g}_i^2 + \sum_{l=1}^d \mathbf{g}_l^2}{\epsilon^2 n^2}$$

which converges to 0 as $n \to \infty$. □

**Proof of Theorem 4.3**

*Proof.*

$$p\left(\hat{\boldsymbol{\eta}}_{i_1} - \hat{\boldsymbol{\eta}}_{i_2} > 0\right) = p_{Z \sim \mathcal{N}(0,1)}\left(Z > -\frac{\boldsymbol{\eta}_{i_1} - \boldsymbol{\eta}_{i_2}}{\sqrt{\left(\frac{\sum_{j \in \{j^{(i_1)}\}} \sigma_j^2 \boldsymbol{\theta}_j^2}{2} + \frac{\sum_{j \in \{j^{(i_2)}\}} \sigma_j^2 \boldsymbol{\theta}_j^2}{2}\right)/2}}\right) \tag{16}$$

$$\sum_{j=1}^{d_i} \sigma_j^2 \boldsymbol{\theta}_j^2 = \sum_{j=1}^{d_i} \frac{\mathbf{g}_j^2 + \sum_{l=1}^d \mathbf{g}_l^2}{nk} \boldsymbol{\theta}_j^2 \leq \frac{\sum_{j=1}^{d_i} \mathbf{g}_j^2 + d_i \sum_{l=1}^d \mathbf{g}_l^2}{nk} u_{\boldsymbol{\theta}}^2 \overset{(*)}{\approx} \frac{d_i u_{\boldsymbol{\theta}}^2 \sum_{l=1}^d \mathbf{g}_l^2}{nk} \tag{17}$$

$$\implies \left(\frac{\sum_{j \in \{j^{(i_1)}\}} \sigma_j^2 \boldsymbol{\theta}_j^2}{2} + \frac{\sum_{j \in \{j^{(i_2)}\}} \sigma_j^2 \boldsymbol{\theta}_j^2}{2}\right)/2 \leq \frac{(d_{i_1} + d_{i_2}) u_{\boldsymbol{\theta}}^2 \sum_{l=1}^d \mathbf{g}_l^2}{4nk} \leq \frac{(d_{i_1} + d_{i_2}) u_{\boldsymbol{\theta}}^2 u_{\mathbf{g}}^2}{4nk} \tag{18}$$

$$\implies p\left(\hat{\boldsymbol{\eta}}_{i_1} - \hat{\boldsymbol{\eta}}_{i_2} > 0\right) \geq \left(Z > -\frac{2\sqrt{nk}(\boldsymbol{\eta}_{i_1} - \boldsymbol{\eta}_{i_2})}{\sqrt{(d_{i_1} + d_{i_2}) u_{\boldsymbol{\theta}}^2 u_{\mathbf{g}}^2}}\right) \overset{(**)}{\geq} 1 - \Phi\left(-\frac{2\sqrt{nk}(q_2 - q_1)/100}{\alpha\sqrt{(d_{i_1} + d_{i_2}) u_{\boldsymbol{\theta}}^2 u_{\mathbf{g}}^2}}\right). \tag{19}$$

(*): $d_i$ is the dimensionality of neuron $i$ and $d$ is the number of parameters, so we have $d \gg d_i$ and, further, $d_i \sum_{l=1}^d \mathbf{g}_l^2 \gg \sum_{j=1}^{d_i} \mathbf{g}_j^2$.

(**): $\boldsymbol{\eta}_i$ is $\alpha$-smooth in $[0, u_{\boldsymbol{\eta}}]$, so $p_{\boldsymbol{\eta}_i}\left(\boldsymbol{\eta}_{i_2} \leq \boldsymbol{\eta}_i \leq \boldsymbol{\eta}_{i_1}\right) = \frac{q_2 - q_1}{100} \leq \alpha(\boldsymbol{\eta}_{i_1} - \boldsymbol{\eta}_{i_2})$, implying that $\boldsymbol{\eta}_{i_1} - \boldsymbol{\eta}_{i_2} \geq \frac{(q_2 - q_1)/100}{\alpha}$. $\qquad\square$

**Proof of Theorem 4.4**

*Proof.* We will argue that for any $i \leq i_\xi(1 - \frac{\epsilon_\xi}{2})$, we have $p(X_i = 1) \geq 1 - \frac{\epsilon_\xi}{2}$. This implies that

$$\sum_{i=1}^{i_\xi} p_i \geq i_\xi \left(1 - \frac{\epsilon_\xi}{2}\right)^2 \geq i_\xi(1 - \epsilon_\xi), \tag{20}$$

as desired. Let us thus focus on $p(X_i = 1)$ for some $i \leq i_\xi(1 - \frac{\epsilon_\xi}{2})$. Using the lower bound on $p_i$ from equation 12, it suffices to prove that

$$p(\hat{\eta}_i > \hat{\eta}_{i_\xi}) \geq 1 - \frac{\epsilon_\xi}{2d_{\boldsymbol{\eta}}}.$$

Thus using Theorem 4.3, it suffices to prove that

$$\Phi\left(-\frac{2c_{d_{\boldsymbol{\eta}}}\alpha\sqrt{(d_i + d_{i_\xi})u_{\boldsymbol{\theta}}^2 u_{\mathbf{g}}^2}(\epsilon_\xi/2)/100}{\alpha\sqrt{(d_i + d_{i_\xi})u_{\boldsymbol{\theta}}^2 u_{\mathbf{g}}^2}}\right) = \Phi\left(-\frac{c_{d_{\boldsymbol{\eta}}}\epsilon_\xi}{100}\right) \leq \frac{\epsilon_\xi}{2d_{\boldsymbol{\eta}}}. \tag{21}$$

Given the fact that $\Phi\left(-\frac{c_{d_{\boldsymbol{\eta}}}\epsilon_\xi}{100}\right) = p_{Z \sim \mathcal{N}(0,1)}\left(Z \leq -\frac{c_{d_{\boldsymbol{\eta}}}\epsilon_\xi}{100}\right)$, we have the following statement: *the greater the $c_{d_{\boldsymbol{\eta}}}$ is, the smaller the $\Phi\left(-\frac{c_{d_{\boldsymbol{\eta}}}\epsilon_\xi}{100}\right)$ is.* This concludes that, there must exist a constant $C_{d_{\boldsymbol{\eta}}}$ such that $\Phi\left(-\frac{c_{d_{\boldsymbol{\eta}}}\epsilon_\xi}{100}\right)$ satisfies Equation 21 $\forall c_{d_{\boldsymbol{\eta}}} \geq C_{d_{\boldsymbol{\eta}}}$. $\qquad\square$

## C Parameter Dependency

Dependencies of parameters between neurons or channels across different layers exist in NNs. These include basic layer connections, residual connections, tensor concatenations, summations, and more, as shown in Figure 3. The black neurons connected by real lines represent the dependent parameters that are in the

same group. Pruning any black neurons results in removing the parameters connected by the real lines. Liu et al. (2021) introduced a group pruning method for CNN models that treats residual connections as grouped dependencies, evaluating and pruning related channels within the same group simultaneously. Similarly, Fang et al. (2023) proposed a novel group pruning technique named Torch-Pruning, which considers various types of dependencies and achieves state-of-the-art results. Ma et al. (2023) further applied this procedure to pruning LLMs. Torch-Pruning can be applied to prune a wide range of neural networks, including image transformers, LLMs, CNNs, and more, making it a popular toolkit for neural network pruning.

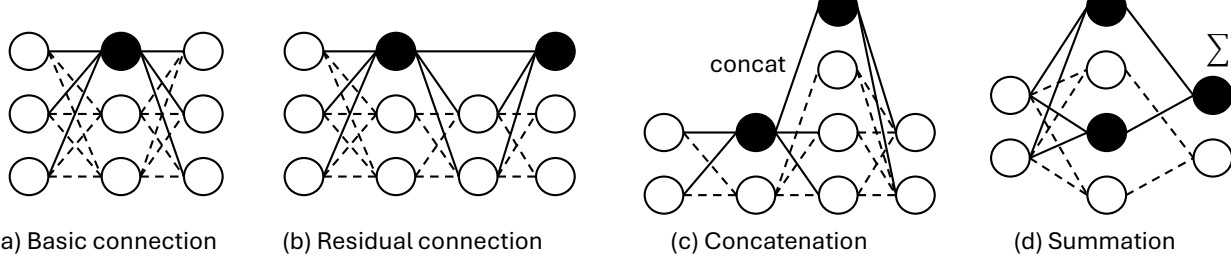

(a) Basic connection      (b) Residual connection      (c) Concatenation      (d) Summation

Figure 3: Common dependencies of parameters in neural networks.

In this study, we also evaluate the influences of incorporating parameter dependency in our approach. We put the experimental results of whether incorporating parameter dependency in Appendix D.2. In the experiments, parameter dependency becomes the following process for our approach: first, searching for dependencies by tracing the computation graph of gradient; next, evaluating the importance of parameter groups; and finally, fine-tuning the parameters within those important groups collectively. For instance, if $\mathbf{W}^{(a)}_{\cdot j}$ and $\mathbf{W}^{(b)}_{i \cdot}$ are dependent, where $\mathbf{W}^{(a)}_{\cdot j}$ is the $j$-th column in parameter matrix (or the $j$-th input channels/features) of layer $a$ and $\mathbf{W}^{(b)}_{i \cdot}$ is the $i$-th row in parameter matrix (or the $i$-th output channels/features) of layer $b$, then $\mathbf{W}^{(a)}_{\cdot j}$ and $\mathbf{W}^{(b)}_{i \cdot}$ will be fine-tuned simultaneously while the corresponding $\mathbf{M}^{(a)}_{dep}$ for $\mathbf{W}^{(a)}_{\cdot j}$ becomes column selection matrix and $\boldsymbol{\delta}^{(a)}$ becomes $\Delta \mathbf{W}^{(a)}_{dep} \mathbf{M}^{(a)}_{dep}$. Consequently, fine-tuning 2.5% output channels for layer $b$ will result in fine-tuning additional 2.5% input channels in each dependent layer. Therefore, for the 5% of desired fine-tuning ratio, the fine-tuning ratio with considering dependencies is set to 2.5%[10] for the approach that includes dependencies.

The forward function of layer $a$ for column selection mentioned above can be written as the following equation:

$$f(\hat{\mathbf{W}}^{(a)}, \mathbf{x}) = f(\mathbf{W}^{(a)}, \mathbf{x}) + f(\mathbf{M}^{(a)} \Delta \mathbf{W}^{(a)}, \mathbf{x}) + f(\Delta \mathbf{W}^{(a)}_{dep} \mathbf{M}^{(a)}_{dep}, \mathbf{x}).$$

Note that in this example, as the dependency is connection between the output feature/channel of $b$ and the input feature/channel of $a$, the dimension $d^{(a)}_{in}$ is equal to $d^{(b)}_{out}$ where $\mathbf{W}^{(a)} \in \mathbb{R}^{d^{(a)}_{out} \times d^{(a)}_{in}}$, $\mathbf{W}^{(b)} \in \mathbb{R}^{d^{(b)}_{out} \times d^{(b)}_{in}}$.

# D    Ablation Studies and Related Analysis

In this section, we first discuss the hyperparameter settings. While we do not include DeBERTaV3 (He et al., 2023) in the main context, we fine-tune DeBERTaV3-base (He et al., 2023) on GLUE. The learning rate is set to $2 \cdot 10^{-5}$ with linear decay, where the decay rate is 0.01. The model is fine-tuned on the full training split of 8 tasks from the GLUE benchmark. The maximum sequence length is fixed to 256 with longer sequences truncated and shorter sequences padded. Note that memory efficiency is not emphasized for small-scale models, as dataset-related memory—particularly with large batch sizes—dominates consumption in these cases. The main advantage of our method in these cases is the reduced FLOPs due to fewer trainable parameters.

---

[10]In some complex models, considering dependencies results in slightly more than twice the number of trainable parameters. However, in most cases, the factor is 2.

Following this, we discuss the computational resource requirements for fine-tuning. Figure 5 illustrates the computation and cache requirements during backpropagation. Next, we provide an ablation study on the impact of different rank settings for our approach and LoRA, as shown in Table 16. Finally, Table 19 demonstrates the advantages of freezing self-attention blocks to reduce memory usage while maintaining performance.

Table 11: Fine-tuning on CIFAR100 and Tiny-ImageNet. #ep and #param represent the number of epochs and the number of trainable parameters, where SPruFT is our method with Taylor importance. Full and Head indicate full fine-tuning and head-finetuning, which only fine-tunes the classification layer. All reported losses and accuracies are based on validation results. **Bold** denotes the best results of each fine-tuning approach (in the same column) on the same model and dataset.

| | CIFAR100 | | | Tiny-ImageNet | | | Caltech101 | | |
|---|---|---|---|---|---|---|---|---|---|
| | Full | Head | SPruFT | Full | Head | SPruFT | Full | Head | SPruFT |
| #ep | loss, acc | loss, acc | loss, acc | loss, acc | loss, acc | loss, acc | loss, acc | loss, acc | loss, acc |
| | DeiT | | | DeiT | | | DeiT | | |
| #param: | 86.0M | 0.2M | 4.6M | 86.1M | 0.3M | 4.8M | 86.0M | 0.2M | 4.6M |
| 5 | **0.36, 90.18** | 0.76, 80.25 | **0.37, 88.70** | **0.54, 87.55** | 0.60, 85.09 | **0.40, 89.69** | 0.11, 97.33 | 1.09, 89.02 | 0.30, 95.41 |
| 10 | 0.44, 90.04 | 0.64, 81.83 | 0.42, 88.62 | 0.69, 86.32 | 0.54, 85.72 | 0.49, 88.96 | **0.11, 97.55** | 0.53, 93.22 | 0.17, 96.28 |
| 30 | 0.62, 89.03 | **0.55, 83.42** | 0.64, 88.61 | 0.94, 84.27 | **0.52, 86.06** | 0.72, 88.67 | 0.11, 97.11 | **0.22, 95.06** | **0.12, 96.50** |
| | ViT | | | ViT | | | ViT | | |
| #param: | 85.9M | 0.1M | 4.5M | 86.0M | 0.2M | 4.6M | 85.9M | 0.1M | 45.2M |
| 5 | **0.38, 90.13** | 1.01, 74.78 | **0.40, 88.13** | **0.51, 88.45** | 0.65, 84.10 | **0.36, 90.87** | 0.12, 97.16 | 1.60, 85.70 | 0.43, 93.96 |
| 10 | 0.45, 89.85 | 0.85, 77.05 | 0.45, 87.55 | 0.66, 86.78 | 0.58, 84.95 | 0.44, 90.48 | **0.11, 97.20** | 0.85, 89.98 | 0.23, 95.54 |
| 30 | 0.62, 88.78 | **0.71, 79.51** | 0.69, 87.83 | 0.96, 84.20 | **0.55, 85.49** | 0.61, 90.56 | 0.12, **97.24** | **0.33, 92.65** | **0.16, 96.02** |
| | ResNet101 | | | ResNet101 | | | ResNet101 | | |
| #param: | 42.7M | 0.2M | 2.2M | 42.9M | 0.4M | 2.4M | 42.7M | 0.2M | 2.2M |
| 5 | **0.50, 86.21** | 1.62, 60.78 | **0.59, 82.36** | **0.92, 77.78** | 1.64, 62.06 | **0.76, 79.66** | 0.14, 96.50 | 1.25, 82.33 | 0.48, 92.56 |
| 10 | 0.58, **86.41** | 1.39, 63.06 | 0.60, 82.33 | 1.10, 76.81 | 1.50, 63.19 | 0.79, 79.54 | **0.14, 96.54** | 0.69, 90.24 | 0.23, 95.58 |
| 30 | 0.80, 84.72 | **1.21, 65.63** | 0.80, **82.49** | 1.54, 74.09 | **1.43, 64.47** | 1.08, 78.58 | 0.18, 95.80 | **0.31, 93.00** | **0.16, 95.89** |
| | ResNeXt101 | | | ResNeXt101 | | | ResNeXt101 | | |
| #param: | 87.0M | 0.2M | 4.9M | 87.2M | 0.4M | 5.1M | 87.0M | 0.2M | 4.9M |
| 5 | **0.47, 87.30** | 1.42, 65.07 | **0.47, 85.94** | **0.86, 79.51** | 1.46, 65.59 | **0.61, 83.88** | **0.12, 97.07** | 1.25, 83.16 | 0.28, 95.84 |
| 10 | 0.56, 87.17 | 1.23, 67.55 | 0.53, 86.04 | 1.01, 79.27 | 1.35, 66.73 | 0.69, 83.47 | 0.13, 96.89 | 0.68, 90.94 | 0.18, 96.28 |
| 30 | 0.71, 86.59 | **1.08, 69.45** | 0.69, **86.33** | 1.41, 76.55 | **1.29, 67.93** | 0.90, 82.83 | 0.16, 96.63 | **0.31, 92.87** | **0.14, 96.76** |

## D.1 Hyperparameter Settings

We report the results of three approaches over several epochs as table 11 and table 12. Overall, full fine-tuning over higher epochs is more prone to overfitting, while head fine-tuning shows the exact opposite trend. Except for the results on caltech101[11], the loss patterns across all models consistently reflect this trend, and most accuracy results further support this conclusion. However, our approach demonstrates a crucial advantage by effectively balancing the tradeoff between performance and computational resources.

Table 11 clearly shows that both our approach and full fine-tuning achieve optimal results within a few epochs, while head fine-tuning requires more training. Notably, all models have been pre-trained on ImageNet-1k, which may explain the strong performance observed with head fine-tuning on Tiny-ImageNet. However, even with this advantage, full fine-tuning still outperforms head fine-tuning, and our approach surpasses both. In just 5 epochs, our approach achieves results comparable to full fine-tuning on all datasets with significantly lower trainable parameters.

In contrast to Table 11, the results in Table 12 show more variation. Although the validation loss follows a similar trend, we report only the evaluation metrics due to the different patterns observed in these metrics. One potential reason for this variation is the varying amounts of training data across the GLUE tasks. As shown in the table, tasks with fewer samples often require more epochs to achieve better performance for both full fine-tuning and our approach. Conversely, for tasks with large amounts of training data such as 'MNLI', 'QNLI', 'QQP', and 'SST-2', the results show tiny improvement from 3 to 10 epochs. Nevertheless, the results still demonstrate that our approach significantly balances the tradeoff between performance

---

[11]The inconsistent trend observed in Caltech101 results is likely due to its significantly smaller sample size.

Table 12: Fine-tuning DeBERTaV3 on GLUE. 'mcc', 'acc', and 'corr' represent 'Matthews correlation', 'accuracy', and 'Pearson correlation', respectively. #param is the number of trainable parameters. SPruFT is our method with Taylor importance, while Full and Head indicate full fine-tuning and head-finetuning, which only fine-tunes the classification layer. All reported metrics are based on validation results, and are percentages. **Bold** denotes the best results of each fine-tuning approach on the same task.

| | | task | CoLA | MNLI | MRPC | QNLI | QQP | RTE | SST-2 | STS-B |
|---|---|---|---|---|---|---|---|---|---|---|
| | | #train | 8.5k | 393k | 3.7k | 108k | 364k | 2.5k | 67k | 7k |
| method | #param | epochs | mcc | acc | acc | acc | acc | acc | acc | corr |
| Full | 184.42M | 3 | **69.96** | **89.42** | 89.71 | **93.57** | **92.08** | 80.14 | **95.53** | 90.44 |
| Full | | 5 | 69.48 | 89.29 | 87.74 | 93.36 | 92.08 | **83.39** | 94.72 | 90.14 |
| Full | | 10 | 68.98 | 88.55 | **90.20** | 93.15 | 91.97 | 80.51 | 93.81 | **90.71** |
| Head | 592.13K | 3 | 24.04 | 62.64 | **68.38** | 70.73 | 80.18 | **52.71** | 65.48 | 5.66 |
| Head | | 5 | 45.39 | 61.75 | **68.38** | **72.32** | 80.59 | 47.29 | **78.44** | 26.88 |
| Head | | 10 | 47.32 | **63.98** | **68.38** | 71.99 | **80.96** | 47.29 | 74.66 | **49.59** |
| SPruFT | 103.57M | 3 | 64.08 | 89.58 | 81.62 | 93.10 | 90.70 | 70.40 | 95.18 | 86.58 |
| SPruFT | | 5 | 65.40 | **90.21** | 86.03 | **93.17** | 90.93 | 74.37 | 95.30 | 87.36 |
| SPruFT | | 10 | **65.56** | 89.55 | **87.50** | 93.15 | **91.57** | **80.14** | **95.41** | **89.14** |

and computational resources. Our method achieves near full fine-tuning performance with remarkably less trainable parameters.

## D.2 Considering Dependency

We evaluate our approach with and without considering parameter dependency, as shown in Table 13 and Table 14.

Table 13: Fine-tuning image models by our SPruFT for 5 epochs. "dep" refers to whether parameter dependencies are involved or not. $\ell^2$, Taylor, and QMTaylor represent the magnitude, Taylor importance, and Quantiles-Mean Taylor importance (Equation 8). Note that QMTaylor is not applied to fine-tuning Caltech101 due to its significantly imbalanced labels. All reported results are validation accuracies. **Bold** indicates the superior results achieved through dependency searching compared to not searching. Underline highlights the best fine-tuning results.

| | data | | CIFAR100 | | | Tiny-ImageNet | | | Caltech101 | |
|---|---|---|---|---|---|---|---|---|---|---|
| model | dep | | $\ell^2$ | Taylor | QMTaylor | $\ell^2$ | Taylor | QMTaylor | $\ell^2$ | Taylor |
| DeiT | ✗ | | **88.05** | **88.70** | **89.37** | **89.31** | **89.69** | **89.75** | **95.01** | **95.41** |
| | ✓ | | 86.43 | 87.33 | 88.08 | 85.56 | 85.92 | 86.49 | 65.35 | 78.04 |
| ViT | ✗ | | **87.13** | **88.06** | **88.51** | **90.78** | **90.87** | **90.90** | **92.69** | **93.96** |
| | ✓ | | 85.24 | 86.83 | 87.91 | 88.83 | 88.95 | 89.67 | 56.30 | 77.82 |
| RN | ✗ | | **82.25** | **82.36** | **83.50** | **79.83** | **79.66** | **80.02** | **93.13** | **92.56** |
| | ✓ | | 78.63 | 78.62 | 81.18 | 69.87 | 69.24 | 72.51 | 54.68 | 52.71 |
| RNX | ✗ | | **86.12** | **85.94** | **86.93** | **83.88** | **83.88** | **84.17** | **95.71** | **95.84** |
| | ✓ | | 84.71 | 85.01 | 85.48 | 79.39 | 78.95 | 79.54 | 92.13 | 91.82 |

We utilize various importance metrics to fine-tune both models using our approach, with and without incorporating parameter dependencies, and report the results to compare their performances. Searching for dependencies in structured pruning is natural, as dependent parameters are pruned together. However, important neurons in a given layer do not always have dependent neurons that are also important in their respective layers. As demonstrated in Table 13, fine-tuning without considering parameter dependencies outperforms fine-tuning incorporating dependencies in all cases. For importance metrics, although the differences between them are not substantial, all results consistently conclude that the Quantile-Mean Taylor importance demonstrates a slight improvement over the standard Taylor importance. Furthermore, both the Quantile-Mean Taylor and standard Taylor metrics outperform the magnitude importance.

Table 14: Fine-tuning DeBERTaV3 on GLUE by our SPSFT for 10 epochs. "dep" refers to whether parameter dependencies are involved or not. Taylor and $\ell^2$ indicate the magnitude and Taylor importance. The importance score is Taylor. We do not apply QMTaylor since the number of labels is tiny. 'mcc', 'acc', and 'corr' represent 'Matthews correlation', 'accuracy', and 'Pearson correlation', respectively. All reported metrics are based on validation results. **Bold** indicates the best results of whether considering dependencies.

| | task | CoLA | MNLI | MRPC | QNLI | QQP | RTE | SST-2 | STS-B |
|---|---|---|---|---|---|---|---|---|---|
| imp | dep | mcc | acc | acc | acc | acc | acc | acc | corr |
| Taylor | ✗ | 65.56 | 89.55 | **87.50** | 93.15 | **91.57** | **80.14** | **95.41** | 89.14 |
| | ✓ | **67.49** | **89.85** | 87.25 | **93.30** | 91.63 | 79.42 | 95.07 | **89.98** |
| $\ell^2$ | ✗ | 65.40 | 89.77 | 83.33 | 92.64 | 91.34 | 74.73 | 94.04 | **88.69** |
| | ✓ | **66.80** | **90.22** | **84.07** | **93.94** | **91.57** | **79.06** | **95.07** | 87.39 |

Table 14 suggests a slightly different conclusion: the impact of parameter dependencies on performance is minor, nearly negligible[12]. However, searching for dependencies involves additional implementations and computational overhead. Combining the results of image models, the conclusion is not searching for the parameter dependencies. For importance metrics, this experiment shows that magnitude and Taylor importance perform similarly.

### D.3 Memory Measurement

In this study, we detail the memory measurement methodology employed. The total memory requirements can be categorized into three main components:

$$\mathrm{mem_{TTL}} = \mathrm{mem_M} + \mathrm{mem_{FT}} + \mathrm{mem_{Aux}},$$

where:

1. $\mathrm{mem_{TTL}}$ is the total memory consumed during training.

2. $\mathrm{mem_M}$ represents the memory consumed by the base model itself.

3. $\mathrm{mem_{FT}}$ corresponds to the memory required for the fine-tuning parameters and their gradients.

4. $\mathrm{mem_{Aux}}$ accounts for any additional memory usage, including optimizer states, caching, and other intermediate computations.

We yield $\mathrm{mem_M}$ by measuring the memory usage during inference on the training data using the pre-trained model. The combined memory usage of $\mathrm{mem_{FT}}$ and $\mathrm{mem_{Aux}}$ is calculated as the difference between $\mathrm{mem_{TTL}}$ and $\mathrm{mem_{Model}}$. For simplicity, we consistently report $\mathrm{mem_{FT}} + \mathrm{mem_{Aux}}$ as "mem" in all comparisons presented in this study.

---

[12]The results of using magnitude importance on the RTE task show significant variation, but this is likely due to the small sample size and the hardness of the task, which result in the unstable performances observed in our experiments. Aside from RTE, the results on other tasks are not significantly different.

Table 15: The requirements of computation resources for fine-tuning. 'mem' traces $\text{mem}_{\text{TTL}} - \text{mem}_{\text{M}}$. [‡] indicates that we freeze the layers for queue, key, and value projection. All fine-tuning parameters are stored in full precision. We also examined the training time and observed that DoRA requires 50% to 100% more time than other methods, while LoRA, RoSA, and our approach need similar training time (differing only by a few seconds). However, due to the influence of various factors on training time and the difficulty of ensuring a fair comparison, we chose not to include these results in our report.

| FT setting | Llama2(7B) | | | | Llama3(8B) | | | |
|---|---|---|---|---|---|---|---|---|
| | #param | $\text{mem}_{\text{TTL}}$ | $\text{mem}_{\text{M}}$ | mem | #param | $\text{mem}_{\text{TTL}}$ | $\text{mem}_{\text{M}}$ | mem |
| LoRA, $r = 64$ | 159.9M(2.37%) | 53.33GB | 29.87GB | 23.46GB | 167.8M(2.09%) | 64.23GB | 33.86GB | 30.37GB |
| RoSA, $r = 32, d = 1.2\%$ | 157.7M(2.34%) | 74.56GB | 29.87GB | 44.69GB | 167.6M(2.09%) | 82.26GB | 33.86GB | 48.40GB |
| DoRA, $r = 64$ | 161.3M(2.39%) | 74.72GB | 29.87GB | 44.85GB | 169.1M(2.11%) | 85.31GB | 33.86GB | 51.45GB |
| VeRA, $r = 64$ | 1.37M(0.02%) | 52.84GB | 29.87GB | 22.97GB | 1.39M(0.02%) | 63.35GB | 33.86GB | 29.49GB |
| SPruFT, $r = 128$ | 145.8M(2.16%) | **47.49GB** | 29.87GB | **17.62GB** | 159.4M(1.98%) | **58.35GB** | 33.86GB | **24.49GB** |
| LoRA[‡], $r = 64$ | 109.6M(1.63%) | 48.68GB | 29.87GB | 18.81GB | 130.0M(1.62%) | 59.90GB | 33.86GB | 26.04GB |
| RoSA[‡], $r = 32, d = 1.2\%$ | 113.2M(1.68%) | 69.70GB | 29.87GB | 39.83GB | 139.1M(1.74%) | 77.60GB | 33.86GB | 43.74GB |
| DoRA[‡], $r = 64$ | 110.5M(1.64%) | 64.05GB | 29.87GB | 34.18GB | 131.2M(1.64%) | 77.97GB | 33.86GB | 44.11GB |
| VeRA, $r = 64$ | 0.84M(0.01%) | 47.63GB | 29.87GB | 17.76GB | 1.05M(0.01%) | 59.64GB | 33.86GB | 25.78GB |
| SPruFT[‡], $r = 128$ | 95.4M(1.42%) | **45.45GB** | 29.87GB | **15.58GB** | 109.1M(1.36%) | **56.57GB** | 33.86GB | **22.71GB** |

## D.4 Resource Requirements

Table 15 presents the resource requirements of various PEFT methods. We compare our approach with LoRA and several of its variants that maintain or surpass LoRA's performance. As shown, our method is the most resource-efficient among these approaches. The subsequent ablation study further demonstrates that our approach achieves performance comparable to LoRA. We exclude comparisons with VeRA (Kopiczko et al., 2024), which proposes sharing a single pair of random low-rank matrices across all layers to save memory footprint. While VeRA achieves some memory savings, its performance often deteriorates.

We note that while our approach offers significant memory efficiency, this benefit is less pronounced in small-scale models, where the primary memory consumption arises from the dataset—especially with large batch sizes. The main advantage of our method in these cases is the reduced FLOPs due to fewer trainable parameters. Therefore, we do not highlight memory efficiency in small-scale model scenarios.

In Section 3, we explain that the memory usage of DoRA is significantly higher than that of LoRA due to its complex computation. We demonstrate the computation graph of DoRA here, as shown in Figure 4. DoRA decomposes $\mathbf{W}$ into magnitude $\mathbf{m}$ and direction $\mathbf{V}$ and computes the final parameters matrix by $\mathbf{W}' = \mathbf{m}\frac{\mathbf{V}+\Delta\mathbf{V}}{||\mathbf{V}+\Delta\mathbf{V}||_c}$. This complicated computation significantly increases memory usage because it requires caching a lot of intermediate values for computing gradients of $\mathbf{B}$, $\mathbf{A}$, and $\mathbf{m}$. As illustrated in Figure 4, each node passed by backpropagation stores some intermediate values for efficient gradient computing.

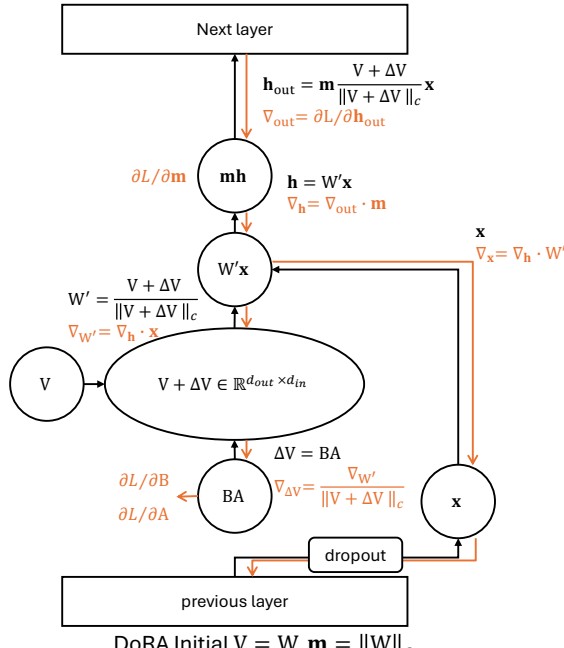

Figure 4: The illustration of DoRA's computation graph. Black operations occur during the forward pass, while orange operations take place during the backward pass.

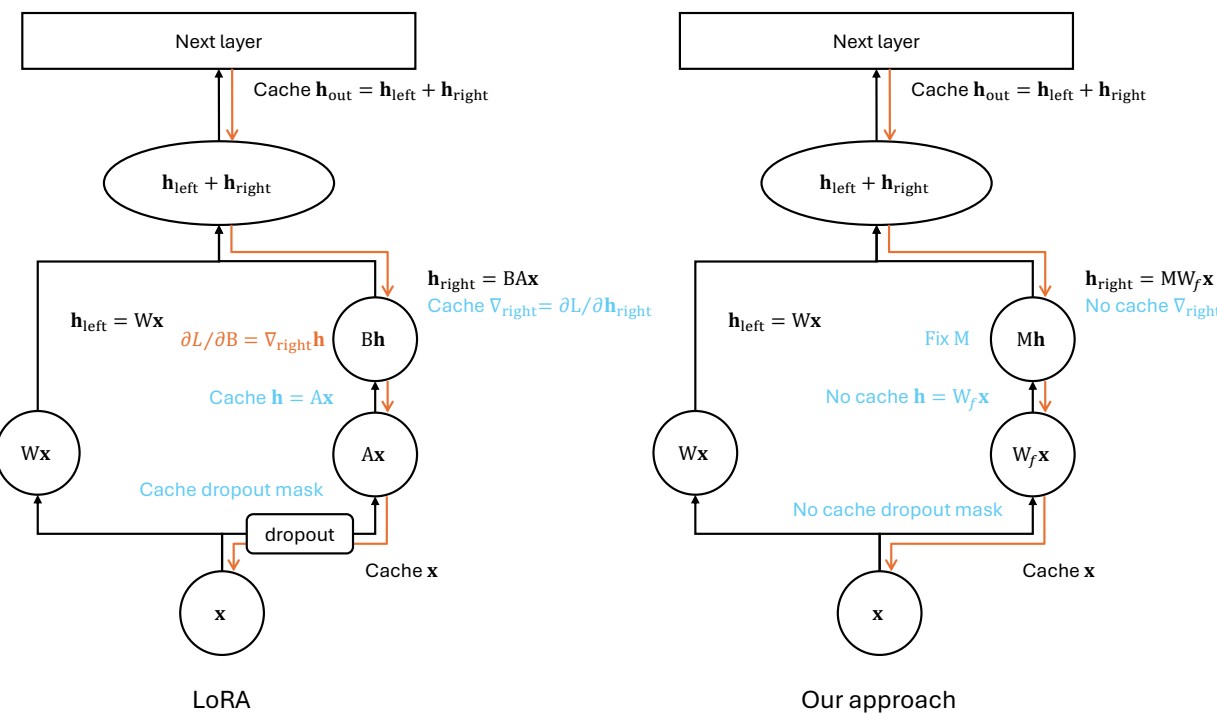

Figure 5: The illustration of backpropagation highlights the operations involved. Black operations occur during the forward pass, while orange operations take place during the backward pass. Blue operations highlight the benefits of our approach. Notably, since $\mathbf{M}$ is non-trainable, caching $\Delta\mathbf{W}\mathbf{x}$ during the forward pass is unnecessary, leading to significant memory savings. Additionally, in practice, PyTorch caches $\frac{\partial L}{\partial \mathbf{h}_{\text{right}}}$ to efficiently compute $\frac{\partial L}{\partial \mathbf{B}}$, although this caching is not strictly required for backpropagation.

### D.5 Cache Benefit

In the main context, we have already shown the memory cost of dropout layers in LoRA, in this section, we will discuss some other benefits of our approach. Figure 5 illustrates the computation and cache requirements in backpropagation (Rumelhart et al., 1986). For simplicity, we replace the notation $f(\cdot, \cdot)$ with different $\mathbf{h}$. With the same number of trainable parameters, our approach eliminates the need to cache $\mathbf{h} = \Delta\mathbf{W}\mathbf{x}$ shown in the figure. While this benefit is negligible under lower rank settings ($r$) or when the number of fine-tuning layers is small, it becomes significant as the model size and rank settings increase. Although the caching requirement for $\mathbf{h}$ can be addressed by recomputing $\mathbf{h} = \mathbf{A}\mathbf{x}$ during backpropagation, this would result in increased time complexity during training.

Table 16: Similar to Table 2, this is the full table with different rank settings for our SPruFT and LoRA. We also present the results of freezing queue-, key-, and value-projection in this table ($\ddagger$)

| Model, ft setting | mem | #param | BoolQ | PIQA | SIQA | HS | WG | ARC-c | ARC-e | OBQA | Avg |
|---|---|---|---|---|---|---|---|---|---|---|---|
| Llama2(7B), LoRA, $r = 8$ | 21.35GB | 20.0M(0.30%) | 80.4 | 73.6 | 70.2 | 79.8 | 63.4 | 68.6 | 80.8 | 71.6 | 73.55 |
| Llama2(7B), SPruFT, $r = 16$ | **15.26GB** | 18.2M(0.27%) | 73.4 | 72.0 | 64.8 | 78.2 | 49.6 | 62.8 | 78.6 | 63.2 | 67.83 |
| Llama2(7B), LoRA, $r = 16$ | 21.64GB | 40.0M(0.59%) | 76.4 | 71.4 | 69.4 | 81.2 | 59.8 | 67.0 | 81.4 | **73.2** | **72.48** |
| Llama2(7B), SPruFT, $r = 32$ | **15.57GB** | 36.4M(0.54%) | **78.2** | **73.6** | 67.4 | 79.4 | 58.8 | **69.0** | **82.2** | 69.4 | 72.25 |
| Llama2(7B), LoRA, $r = 32$ | 22.21GB | 80.0M(1.19%) | 74.8 | **74.2** | 71.4 | 78.4 | 58.6 | 67.0 | **82.2** | 69.6 | 72.03 |
| Llama2(7B), SPruFT, $r = 64$ | **16.20GB** | 72.9M(1.08%) | **78.0** | 73.8 | 66.4 | **83.4** | 59.6 | 69.4 | 79.0 | **71.6** | **72.65** |
| Llama2(7B), LoRA, $r = 64$ | 23.46GB | 159.9M(2.37%) | 77.0 | **76.2** | 67.8 | 84.2 | 62.6 | 70.0 | 82.0 | **74.0** | 74.23 |
| Llama2(7B), SPruFT, $r = 128$ | **17.62GB** | 145.8M(2.16%) | **80.0** | 75.2 | 67.6 | **85.0** | 63.4 | 70.8 | **82.4** | 71.8 | **74.53** |
| Llama2(7B), LoRA, $r = 128$ | 25.98GB | 319.8M(4.75%) | 77.2 | **76.2** | 72.0 | 84.2 | 62.0 | 71.6 | 81.2 | 71.8 | 74.53 |
| Llama2(7B), SPruFT, $r = 256$ | **20.47GB** | 291.5M(4.33%) | **80.6** | 73.0 | 66.6 | 83.8 | 63.2 | 69.6 | **82.4** | 70.2 | 73.68 |
| Llama2(7B), LoRA, $r = 256$ | 31.00GB | 639.6M(9.49%) | 78.0 | **73.6** | 71.2 | 82.4 | 60.8 | 67.8 | **83.0** | 74.4 | 73.90 |
| Llama2(7B), SPruFT, $r = 512$ | **26.17GB** | 583.0M(8.65%) | **79.4** | 73.2 | 68.8 | 81.4 | 62.2 | 69.8 | 81.4 | **75.2** | **73.93** |
| Llama2(7B), LoRA$^{\ddagger}$, $r = 16$ | 17.62GB | 27.4M(0.41%) | 80.4 | **75.6** | 68.2 | 77.4 | 58.0 | 66.4 | 80.2 | **71.8** | **72.25** |
| Llama2(7B), SPruFT$^{\ddagger}$, $r = 32$ | **14.29GB** | 23.9M(0.35%) | **82.6** | 73.0 | 64.4 | **80.0** | 61.4 | 64.8 | **80.4** | 70.2 | 72.10 |
| Llama2(7B), LoRA$^{\ddagger}$, $r = 32$ | 17.95GB | 54.8M(0.81%) | 75.6 | **76.0** | 68.6 | 79.2 | 63.6 | 67.2 | **82.0** | 71.0 | **72.90** |
| Llama2(7B), SPruFT$^{\ddagger}$, $r = 64$ | **14.67GB** | 47.7M(0.71%) | **78.2** | 75.8 | 66.4 | **81.4** | 59.2 | **67.8** | 78.8 | **72.0** | 72.45 |
| Llama2(7B), LoRA$^{\ddagger}$, $r = 64$ | 18.81GB | 109.6M(1.63%) | 79.6 | 73.8 | 67.0 | **80.4** | 62.6 | 69.2 | **81.0** | 70.4 | 73.00 |
| Llama2(7B), SPruFT$^{\ddagger}$, $r = 128$ | **15.58GB** | 95.4M(1.42%) | **81.4** | **75.0** | 67.2 | **80.4** | 65.0 | 69.2 | 79.8 | **73.8** | **73.98** |
| Llama3(8B), LoRA, $r = 8$ | 28.60GB | 21.0M(0.26%) | **84.8** | **79.8** | 74.4 | **88.4** | 70.2 | 79.8 | 90.8 | 78.8 | **80.88** |
| Llama3(8B), SPruFT, $r = 16$ | **22.32GB** | 19.9M(0.25%) | 83.0 | 78.0 | 66.6 | 77.0 | 60.2 | 74.0 | 89.4 | 75.2 | 75.43 |
| Llama3(8B), LoRA, $r = 16$ | 28.86GB | 41.9M(0.52%) | **85.2** | 80.8 | 68.4 | 81.8 | 69.0 | 79.4 | 90.0 | 77.0 | **78.95** |
| Llama3(8B), SPruFT, $r = 32$ | **22.62GB** | 39.8M(0.50%) | 81.2 | **82.2** | 68.2 | **85.4** | 63.4 | 79.4 | 87.6 | **80.8** | 78.53 |
| Llama3(8B), LoRA, $r = 32$ | 29.37GB | 83.9M(1.04%) | **85.2** | 81.8 | 68.2 | **87.8** | 67.0 | 76.4 | **89.2** | 80.4 | **79.50** |
| Llama3(8B), SPruFT, $r = 64$ | **23.23GB** | 79.7M(0.99%) | 83.2 | 80.6 | **69.4** | 86.0 | 60.6 | **78.4** | 83.4 | **81.4** | 77.88 |
| Llama3(8B), LoRA, $r = 64$ | 30.37GB | 167.8M(2.09%) | 84.2 | 77.0 | 63.2 | 84.2 | 67.2 | 76.4 | 88.8 | 71.0 | 76.50 |
| Llama3(8B), SPruFT, $r = 128$ | **24.49GB** | 159.4M(1.98%) | **87.6** | **77.4** | **71.4** | **85.4** | **70.2** | **79.8** | **90.8** | **81.8** | **80.55** |
| Llama3(8B), LoRA, $r = 128$ | 32.35GB | 335.5M(4.18%) | 85.6 | 81.6 | 71.8 | 84.2 | 67.2 | 78.2 | 91.8 | 81.0 | 80.18 |
| Llama3(8B), SPruFT, $r = 256$ | **26.99GB** | 318.8M(3.97%) | **86.4** | 80.2 | 71.8 | **85.4** | 64.2 | 77.8 | 89.4 | 75.2 | 78.80 |
| Llama3(8B), LoRA, $r = 256$ | 36.32GB | 671.1M(8.36%) | 80.2 | 78.6 | 65.8 | 85.8 | 68.6 | 78.2 | 88.4 | 76.2 | 77.73 |
| Llama3(8B), SPruFT, $r = 512$ | **32.00GB** | 637.5M(7.94%) | **83.4** | **81.2** | **70.6** | **86.4** | 67.2 | 72.8 | 86.8 | **81.2** | **78.70** |
| Llama3(8B), LoRA$^{\ddagger}$, $r = 16$ | 24.88GB | 32.5M(0.41%) | **83.8** | **83.6** | **73.8** | **86.6** | **67.4** | 79.6 | **91.2** | 78.6 | **80.58** |
| Llama3(8B), SPruFT$^{\ddagger}$, $r = 32$ | **21.37GB** | 27.3M(0.34%) | 83.2 | 82.6 | 66.6 | 86.0 | 65.6 | **82.4** | 89.8 | **80.4** | 79.58 |
| Llama3(8B), LoRA$^{\ddagger}$, $r = 32$ | 25.28GB | 65.0M(0.81%) | 83.0 | 68.8 | **71.4** | **85.2** | 66.2 | **81.0** | **91.6** | 79.6 | 78.35 |
| Llama3(8B), SPruFT$^{\ddagger}$, $r = 64$ | **21.81GB** | 54.5M(0.68%) | **84.4** | **78.0** | 71.0 | 83.2 | **69.0** | 79.0 | 84.4 | **83.4** | **79.05** |
| Llama3(8B), LoRA$^{\ddagger}$, $r = 64$ | 26.04GB | 130.0M(1.62%) | 86.2 | **81.6** | 67.8 | 81.8 | 66.0 | 73.8 | 86.2 | **78.2** | 77.73 |
| Llama3(8B), SPruFT$^{\ddagger}$, $r = 128$ | **22.71GB** | 109.1M(1.36%) | **89.4** | 79.0 | **71.2** | **86.2** | **71.8** | **83.0** | **86.8** | 76.2 | **80.45** |

### D.6 Rank Settings

We present an ablation study of rank settings here. Table 16 demonstrates that $r = 8$ is sufficient for LoRA when fine-tuning Llama-2 and Llama-3. In contrast, increasing $r$ for our approach yields slight performance improvements. The most remarkable observation in Table 16 is the exceptional memory efficiency of our approach: even with $r = 128$, the memory usage of our method is significantly lower than that of LoRA with $r = 8$.

Table 17 and Table 18 are the full tables of importance evaluation for Llama2 and Llama3.

Table 17: Similar to Table 5, this is the full table with different rank settings. We also present the results of freezing queue-, key-, and value-projection in this table ($^\ddagger$).

| Model, ft setting | BoolQ | PIQA | SIQA | HS | WG | ARC-c | ARC-e | OBQA | Avg |
|---|---|---|---|---|---|---|---|---|---|
| Llama2(7B), SPruFT | | | | | | | | | |
| $r = 16$, random | 72.8 | 72.4 | 63.2 | 73.6 | 51.4 | 61.0 | **78.6** | 61.0 | 66.75 |
| $r = 16$, $\ell^2$ | 73.4 | 72.0 | 64.8 | 78.2 | 49.6 | 62.8 | **78.6** | **63.2** | 67.83 |
| $r = 16$, ZOTaylor | **74.4** | **73.6** | **69.4** | **80.8** | **57.2** | **70.0** | 77.2 | 63.0 | **70.70** |
| $r = 32$, random | 72.2 | 70.8 | 65.6 | 77.0 | 55.4 | 71.0 | 77.0 | 69.2 | 69.80 |
| $r = 32$, $\ell^2$ | **78.2** | **73.6** | **67.4** | 79.4 | 58.8 | 69.0 | **82.2** | 69.4 | 72.25 |
| $r = 32$, ZOTaylor | 74.4 | 72.0 | 66.6 | **80.4** | **63.4** | **71.8** | 76.6 | **73.6** | **72.35** |
| $r = 64$, random | 71.0 | 68.0 | 62.8 | 74.2 | **63.6** | 66.6 | 78.0 | 71.4 | 69.45 |
| $r = 64$, $\ell^2$ | 78.0 | **73.8** | 66.4 | **83.4** | 59.6 | 69.4 | 79.0 | **71.6** | 72.65 |
| $r = 64$, ZOTaylor | **78.2** | 71.0 | **68.4** | 80.8 | 62.0 | **69.6** | **81.0** | 71.4 | **72.80** |
| $r = 128$, random | **80.0** | 70.6 | **69.8** | 68.2 | 57.4 | 67.6 | 74.6 | 70.2 | 69.80 |
| $r = 128$, $\ell^2$ | **80.0** | **75.2** | 67.6 | **85.0** | 63.4 | 70.8 | 82.4 | **71.8** | **74.53** |
| $r = 128$, ZOTaylor | 79.4 | 74.4 | 67.0 | 82.4 | **65.0** | **72.4** | **83.0** | 71.2 | 74.35 |
| $r = 256$, random | 71.0 | 71.2 | **69.8** | 70.4 | 62.4 | 70.4 | 78.4 | 60.2 | 69.23 |
| $r = 256$, $\ell^2$ | **80.6** | 73.0 | 66.6 | 83.8 | 63.2 | 69.6 | **82.4** | 70.2 | 73.68 |
| $r = 256$, ZOTaylor | 78.4 | **73.4** | 63.2 | **83.4** | **66.0** | **71.6** | 81.8 | **73.8** | **73.95** |
| $r = 512$, random | 73.8 | 70.6 | 66.0 | 67.2 | 59.8 | 68.6 | 75.2 | 67.6 | 68.60 |
| $r = 512$, $\ell^2$ | **79.4** | **73.2** | **68.8** | 81.4 | 62.2 | 69.8 | **81.4** | 75.2 | **73.93** |
| $r = 512$, ZOTaylor | 77.4 | **73.2** | 61.6 | **82.2** | **68.2** | **72.2** | 76.4 | **76.4** | 73.45 |
| $r = 32^\ddagger$, random | 78.0 | 71.2 | 64.8 | 72.6 | 55.8 | 67.8 | 76.4 | 69.8 | 69.55 |
| $r = 32^\ddagger$, $\ell^2$ | **82.6** | **73.0** | 64.4 | **80.0** | **61.4** | 64.8 | **80.4** | **70.2** | **72.10** |
| $r = 32^\ddagger$, ZOTaylor | 77.4 | 71.2 | **65.6** | 78.0 | 59.4 | **68.8** | 79.8 | 69.0 | 71.15 |
| $r = 64^\ddagger$, random | 75.8 | 71.0 | 64.4 | 70.0 | 58.8 | **71.2** | 79.8 | **72.0** | 70.38 |
| $r = 64^\ddagger$, $\ell^2$ | 78.2 | **75.8** | 66.4 | 81.4 | 59.2 | 67.8 | 78.8 | **72.0** | 72.45 |
| $r = 64^\ddagger$, ZOTaylor | **81.4** | 71.4 | **67.0** | **82.2** | **61.6** | **71.2** | **80.0** | **72.0** | **73.34** |
| $r = 128^\ddagger$, random | 75.8 | **75.2** | 67.2 | 79.2 | 64.0 | 69.6 | 77.0 | 68.6 | 72.08 |
| $r = 128^\ddagger$, $\ell^2$ | **81.4** | 75.0 | 67.2 | 80.4 | **65.0** | 69.2 | 79.8 | **73.8** | 73.98 |
| $r = 128^\ddagger$, ZOTaylor | 81.0 | 74.4 | **69.0** | **81.0** | 64.6 | **70.6** | **81.2** | 72.8 | **74.30** |
| Llama3(8B), SPruFT | | | | | | | | | |
| $r = 16$, random | **85.4** | 78.0 | **70.8** | 81.6 | **67.6** | 76.8 | **89.4** | 74.4 | 78.00 |
| $r = 16$, $\ell^2$ | 83.0 | 78.0 | 66.6 | 77.0 | 60.2 | 74.0 | **89.4** | 75.2 | 75.43 |
| $r = 16$, ZOTaylor | **85.4** | **82.4** | 68.8 | **85.6** | 62.4 | 74.6 | 89.2 | **78.2** | **78.33** |
| $r = 32$, random | **83.8** | 78.4 | 59.2 | **86.4** | 69.8 | 79.4 | **88.4** | 76.6 | 77.75 |
| $r = 32$, $\ell^2$ | 81.2 | **82.2** | 68.2 | 85.4 | 63.4 | **79.4** | 87.6 | 80.8 | 78.53 |
| $r = 32$, ZOTaylor | 81.6 | 78.4 | **70.4** | 85.0 | 63.2 | 79.2 | 88.2 | **84.6** | **78.83** |
| $r = 64$, random | 82.6 | 73.4 | 68.8 | 72.6 | 63.4 | 74.0 | 81.6 | 75.0 | 73.93 |
| $r = 64$, $\ell^2$ | **83.2** | **80.6** | **69.4** | **86.0** | 60.6 | **78.4** | 83.4 | 81.4 | 77.88 |
| $r = 64$, ZOTaylor | 81.8 | 78.0 | 68.4 | 85.4 | **64.6** | 77.6 | **85.8** | 82.4 | **78.00** |
| $r = 128$, random | 84.2 | 77.4 | 70.2 | 72.0 | **72.4** | 72.8 | 84.0 | 75.6 | 76.08 |
| $r = 128$, $\ell^2$ | 87.6 | 77.4 | **71.4** | 85.4 | 70.2 | 79.8 | 90.8 | 81.8 | 80.55 |
| $r = 128$, ZOTaylor | **89.0** | **78.8** | 70.6 | **86.2** | 69.4 | **80.4** | **92.0** | **83.8** | **81.28** |
| $r = 256$, random | 84.4 | 77.2 | 67.6 | 78.4 | **72.2** | 75.0 | 86.6 | 74.2 | 76.95 |
| $r = 256$, $\ell^2$ | 86.4 | **80.2** | **71.8** | **85.4** | 64.2 | 77.8 | 89.4 | 75.2 | 78.80 |
| $r = 256$, ZOTaylor | **88.4** | 78.2 | 71.6 | 84.2 | 67.2 | **79.2** | **91.4** | **81.4** | **80.20** |
| $r = 512$, random | 83.4 | 73.2 | 66.8 | 78.4 | **67.2** | 74.8 | 86.4 | 75.2 | 75.68 |
| $r = 512$, $\ell^2$ | 83.4 | **81.2** | 70.6 | **86.4** | **67.2** | 72.8 | 86.8 | **81.2** | 78.70 |
| $r = 512$, ZOTaylor | **87.4** | **81.2** | **72.8** | 83.4 | 66.6 | **79.8** | **88.4** | 80.2 | **79.98** |
| $r = 32^\ddagger$, random | 81.6 | 80.0 | 69.2 | **87.6** | 69.8 | 82.4 | 89.6 | 85.0 | **80.65** |
| $r = 32^\ddagger$, $\ell^2$ | 83.2 | **82.6** | 66.6 | 86.0 | 65.6 | 82.4 | 89.8 | 80.4 | 79.58 |
| $r = 32^\ddagger$, ZOTaylor | **86.2** | 77.6 | **72.2** | 84.0 | 65.6 | 78.8 | **90.0** | **86.4** | 80.10 |
| $r = 64^\ddagger$, random | 81.2 | **80.2** | 61.8 | 82.8 | **70.6** | 79.6 | **89.2** | 82.0 | 78.43 |
| $r = 64^\ddagger$, $\ell^2$ | 84.4 | 78.0 | 71.0 | 83.2 | 69.0 | 79.0 | 84.4 | 83.4 | 79.05 |
| $r = 64^\ddagger$, ZOTaylor | **84.6** | 79.0 | **72.4** | **88.0** | 61.2 | 78.4 | 88.8 | **86.8** | **79.90** |
| $r = 128^\ddagger$, random | 82.8 | 75.2 | 68.8 | 83.8 | 67.8 | 78.2 | 83.0 | 78.0 | 77.20 |
| $r = 128^\ddagger$, $\ell^2$ | 89.4 | **79.0** | **71.2** | **86.2** | 71.8 | **83.0** | 86.8 | 76.2 | 80.45 |
| $r = 128^\ddagger$, ZOTaylor | **90.6** | 77.0 | 71.0 | 84.0 | **72.6** | 80.0 | **92.6** | **86.6** | **81.80** |

Table 18: Importance evaluation for Llama2 and Llama3 on MT-Bench. We also present the results of freezing queue-, key-, and value-projection in this table ($^\ddagger$). **Bold** indicates the best result on the same task.

| Model, ft setting | Coding | Extraction | Humanities | Math | Reasoning | Roleplay | Stem | Writing | Avg |
|---|---|---|---|---|---|---|---|---|---|
| Llama2(7B), SPruFT | | | | | | | | | |
| $r = 128$, random | 0.67 | **3.44** | 5.11 | 1.67 | 3.50 | 4.95 | 4.41 | 3.89 | 3.45 |
| $r = 128$, $\ell$ | **1.82** | 2.55 | 4.80 | 2.08 | **4.07** | 4.79 | **4.50** | 3.53 | 3.52 |
| $r = 128$, ZOTaylor | 1.42 | 2.63 | **5.16** | **2.36** | 3.67 | **5.20** | 4.39 | **4.11** | **3.62** |
| $r = 128^\ddagger$, random | 2.31 | **2.90** | 4.75 | **2.77** | 2.94 | 4.61 | 4.35 | **4.15** | 3.60 |
| $r = 128^\ddagger$, $\ell$ | 1.08 | 2.10 | 4.60 | 1.15 | **3.80** | 4.22 | 4.33 | 3.42 | 3.09 |
| $r = 128^\ddagger$, ZOTaylor | **2.44** | 2.63 | **5.20** | 2.21 | 2.93 | **5.00** | **4.94** | 3.95 | **3.66** |
| Llama3(8B), SPruFT | | | | | | | | | |
| $r = 128$, random | 3.22 | 3.74 | 4.72 | 3.33 | 3.50 | 5.20 | 6.38 | 5.11 | 4.40 |
| $r = 128$, $\ell$ | 3.88 | **5.11** | 6.11 | 3.83 | **4.21** | 5.35 | **6.44** | **5.40** | 5.04 |
| $r = 128$, ZOTaylor | **4.13** | 4.78 | **6.89** | **6.33** | 4.08 | **5.95** | 5.39 | 5.16 | **5.34** |
| $r = 128^\ddagger$, random | 3.56 | 5.06 | 5.68 | 4.69 | 4.00 | 5.26 | **6.17** | **6.00** | 5.05 |
| $r = 128^\ddagger$, $\ell$ | **4.75** | 4.78 | 5.16 | 3.67 | 3.31 | **6.25** | 6.06 | 5.00 | 4.87 |
| $r = 128^\ddagger$, ZOTaylor | 4.13 | **5.38** | **6.05** | **4.79** | **5.00** | 5.22 | 5.88 | 5.21 | **5.21** |

Table 19: Same results of Table 16 with a reordering of the rows. This table is for comparing *fine-tuning all linear layers* with *freezing queue-, key-, and value-projection.*

| Model, ft setting | mem | #param | BoolQ | PIQA | SIQA | HS | WG | ARC-c | ARC-e | OBQA | Avg |
|---|---|---|---|---|---|---|---|---|---|---|---|
| Llama2(7B), LoRA, $r = 16$ | 21.64GB | 40.0M(0.59%) | **76.4** | 71.4 | **69.4** | **81.2** | 59.8 | 67.0 | 81.4 | **73.2** | 72.48 |
| Llama2(7B), LoRA$^\ddagger$, $r = 32$ | **17.95GB** | 54.8M(0.81%) | 75.6 | **76.0** | 68.6 | 79.2 | **63.6** | **67.2** | **82.0** | 71.0 | **72.90** |
| Llama2(7B), LoRA, $r = 32$ | 22.21GB | 80.0M(1.19%) | 74.8 | **74.2** | **71.4** | 78.4 | 58.6 | 67.0 | **82.2** | 69.6 | 72.03 |
| Llama2(7B), LoRA$^\ddagger$, $r = 64$ | **18.81GB** | 109.6M(1.63%) | **79.6** | 73.8 | 67.0 | **80.4** | **62.6** | **69.2** | 81.0 | **70.4** | **73.00** |
| Llama2(7B), SPruFT, $r = 32$ | 15.57GB | 36.4M(0.54%) | **78.2** | 73.6 | **67.4** | 79.4 | 58.8 | **69.0** | **82.2** | 69.4 | 72.25 |
| Llama2(7B), SPruFT$^\ddagger$, $r = 64$ | **14.67GB** | 47.7M(0.71%) | **78.2** | **75.8** | 66.4 | **81.4** | **59.2** | 67.8 | 78.8 | **72.0** | **72.45** |
| Llama2(7B), SPruFT, $r = 64$ | 16.20GB | 72.9M(1.08%) | 78.0 | 73.8 | 66.4 | **83.4** | 59.6 | **69.4** | 79.0 | 71.6 | 72.65 |
| Llama2(7B), SPruFT$^\ddagger$, $r = 128$ | **15.58GB** | 95.4M(1.42%) | **81.4** | **75.0** | **67.2** | 80.4 | **65.0** | 69.2 | **79.8** | **73.8** | **73.98** |
| Llama3(8B), LoRA, $r = 16$ | 28.86GB | 41.9M(0.52%) | **85.2** | **80.8** | 68.4 | 81.8 | **69.0** | 79.4 | 90.0 | 77.0 | **78.95** |
| Llama3(8B), LoRA$^\ddagger$, $r = 32$ | **25.28GB** | 65.0M(0.81%) | 83.0 | 68.8 | **71.4** | **85.2** | 66.2 | **81.0** | **91.6** | **79.6** | 78.35 |
| Llama3(8B), LoRA, $r = 32$ | 29.37GB | 83.9M(1.04%) | 85.2 | **81.8** | **68.2** | **87.8** | **67.0** | 76.4 | **89.2** | **80.4** | **79.50** |
| Llama3(8B), LoRA$^\ddagger$, $r = 64$ | **26.04GB** | 130.0M(1.62%) | **86.2** | 81.6 | 67.8 | 81.8 | 66.0 | 73.8 | 86.2 | 78.2 | 77.73 |
| Llama3(8B), SPruFT, $r = 32$ | 22.62GB | 39.8M(0.50%) | 81.2 | **82.2** | 68.2 | **85.4** | 63.4 | **79.4** | **87.6** | 80.8 | 78.53 |
| Llama3(8B), SPruFT$^\ddagger$, $r = 64$ | **21.81GB** | 54.5M(0.68%) | **84.4** | 78.0 | **71.0** | 83.2 | **69.0** | 79.0 | 84.4 | **83.4** | **79.05** |
| Llama3(8B), SPruFT, $r = 64$ | 23.23GB | 79.7M(0.99%) | 83.2 | **80.6** | 69.4 | 86.0 | 60.6 | 78.4 | 83.4 | **81.4** | 77.88 |
| Llama3(8B), SPruFT$^\ddagger$, $r = 128$ | **22.71GB** | 109.1M(1.36%) | **89.4** | 79.0 | **71.2** | **86.2** | **71.8** | **83.0** | **86.8** | 76.2 | **80.45** |

### D.7 Benefit of Freezing Attention Blocks

We now assess different fine-tuning strategies. Table 19 highlights the importance of selecting fine-tuning layers strategically to minimize redundant memory usage. Freezing the self-attention blocks achieves performance comparable to fine-tuning all layers while significantly reducing memory consumption during training. This efficiency stems from reducing the need to cache intermediate outputs for gradient computation. For example, as illustrated in Figure 5, using LoRA, $\nabla_{out}$ must be cached to compute $\frac{\partial L}{\partial A}$ for the subsequent layer. Freezing the next layer eliminates this caching requirement, further optimizing memory usage.

## E Details of Datasets

### E.1 Vision Benchmarks

**CIFAR100**: CIFAR100 (Krizhevsky et al., 2009) has 100 classes with 600 images of size 32x32 per class, while the CIFAR10 has 10 classes with 6000 images per class. In this study, we use the CIFAR100 downloaded from huggingface (`https://huggingface.co/datasets/uoft-cs/cifar100`) with 500 training images and 100 validation images per class. In our experiments, we resize the images to 256x256, crop the center to 224x224, and normalize them using the CIFAR mean $(0.507, 0.487, 0.441)$ and standard deviation $(0.267, 0.256, 0.276)$ for the three channels.

**Tiny-ImageNet**: Tiny-ImageNet (Tavanaei, 2020) has 200 classes with images of size 64x64, while the full ImageNet-1k (Deng et al., 2009) has all 1000 classes where each image is the standard size 224x224. In this study, we use the Tiny-ImageNet downloaded from huggingface (`https://huggingface.co/datasets/zh-plus/tiny-imagenet`) with 500 training images and 50 validation images per class. In our experiments, we resize the images to 256x256, crop the center to 224x224, and normalize them using the mean $(0.485, 0.456, 0.406)$ and standard deviation $(0.229, 0.224, 0.225)$ for the three channels.

**caltech101**: Caltech101 (Li et al., 2022) consists of 101 classes, with images of varying sizes typically having edge lengths between 200 and 300 pixels. Each class contains approximately 40 to 800 images, resulting in a total of around 9,000 images. In this study, we use the Caltech101 dataset provided by PyTorch (`https://pytorch.org/vision/main/generated/torchvision.datasets.Caltech101.html`), allocating 75% of the images for training and the remaining 25% for validation. In our experiments, we preprocess the images by resizing them to 256×256, cropping the center to 224×224, and normalizing them using the mean $(0.485, 0.456, 0.406)$ and standard deviation $(0.229, 0.224, 0.225)$ for the three channels.

### E.2 General Language Understanding Evaluation Benchmark (GLUE)

**CoLA**: The Corpus of Linguistic Acceptability (CoLA) is a dataset for assessing linguistic acceptability (Warstadt et al., 2018). This task is a binary classification for predicting whether a sentence is grammatically acceptable. The dataset is primarily from books and journal articles on linguistic theory.

**MNLI**: The Multi-Genre Natural Language Inference (MultiNLI) is a dataset designed to evaluate a model's ability to perform natural language inference (NLI). The task is to predict whether the premise entails the hypothesis, contradicts the hypothesis, or neither. The data set contains 433k sentence pairs annotated with textual entailment information (Williams et al., 2018).

**MRPC**: The Microsoft Research Paraphrase Corpus (Dolan & Brockett, 2005) is a dataset designed for evaluating paraphrase detection systems. It consists of sentence pairs, with binary labels of whether the two sentences in the pair are equivalent. The data are automatically extracted from online news and labeled by humans.

**QNLI**: The Stanford Question Answering Dataset (SQuAD) is a dataset designed for machine comprehension of text (Rajpurkar et al., 2016). The dataset consists of question-paragraph pairs, where one of the sentences in the paragraph contains the answer to the corresponding question. The paragraphs are from Wikipedia and the questions are written by human annotators.

**QQP**: The Quora Question Pairs (QQP) dataset is a dataset of question pairs (`https://data.quora.com/First-Quora-Dataset-Release-Question-Pairs`). The task is to determine whether two questions are semantically equivalent.

**RTE**: The Recognizing Textual Entailment (RTE) datasets are a series of challenges that evaluate models' ability to determine whether a premise can entail a given hypothesis (Dagan et al., 2006; Haim et al., 2006; Giampiccolo et al., 2007; Bentivogli et al., 2009). The data are constructed based on the texts from Wikipedia and news. The datasets have been used to evaluate the performance of both traditional language models and the state-of-the-art LLMs.

**SST-2**: The Stanford Sentiment Treebank is a dataset of sentences extracted from movie reviews (Socher et al., 2013). Each sentence is labeled as either positive or negative. The task is to predict whether the sentence is positive or negative.

**STS-B**: The Semantic Textual Similarity Benchmark (STSB) is a dataset with sentence pairs collected from news headlines, video and image captions, and natural language inference data (Cer et al., 2017). The task is to predict the semantic similarity between pairs of sentences. Each pair of sentences is annotated with a similarity score ranging from 0 to 5, where 0 indicates no semantic similarity and 5 indicates semantically equivalent.

### E.3 Text-Generation Datasets

**GSM8k**: GSM8K (Grade School Math 8K) is a dataset of 8792 high-quality grade school math problems, including problems in diverse languages. These problems take between 2 and 8 steps of elementary calculations using basic arithmetic operations ($+ - \times \div$) to solve. The dataset was created to support the task of question answering on basic mathematical problems to evaluate the model's ability of basic arithmetic reasoning.

**Stanford Alpaca**: Alpaca is an instruction dataset designed for instruction training of pre-trained language models (Taori et al., 2023). It contains 52002 instruction-response pairs generated by OpenAI's text-davinci-003 engine or written by humans. Note that there is only a training split in this dataset. Models fine-tuned on Alpaca are often evaluated by other tasks like "EleutherAI LM Harness". Alpaca-GPT is an updated version with the answers generated by GPT-4 (Achiam et al., 2023).

**ARC**: The AI2 Reasoning Challenge (ARC) dataset consists of grade-school level, multiple-choice science questions (Clark et al., 2018). ARC dataset includes a Challenge Set and an Easy Set. The easy set contains questions that can be answered with straightforward reasoning, while the challenge set requires deeper understanding and more reasoning skills. The ARC-Easy includes 2251 training samples, 570 validation samples, and 2376 test samples and the ARC-Challenge includes 1119 training samples, 299 validation samples, and 1172 test samples.

**BoolQ**: Boolean Questions (BoolQ) is a dataset of yes/no question answering (Clark et al., 2019) and includes 9427 training samples and 3270 validation samples. The dataset is designed to assess models' comprehension and reasoning abilities. Each example contains question, passage, answer, and title.

**HellaSwag**: HellaSwag is a dataset designed to evaluate the models' abilities in generating reasonable contexts (Zellers et al., 2019). It consists of prompts with a short context followed by multiple possible continuations. The goal is to find the correct or most plausible option. The training set, validation set, and test set have 39905 samples, 10042 samples, 10003 samples, respectively.

**OpenBookQA**: OpenBookQA is a question-answering dataset (Mihaylov et al., 2018) comprising 4957 training samples, 500 validation samples, and 500 test samples. It requires reasoning ability and a deeper understanding of common knowledge to answer questions. Each data contains a short passage with multiple possible answers. The dataset emphasizes the integration of world knowledge and reasoning skills, making it a challenging benchmark for natural language processing models. It tests models' abilities to understand and apply factual information effectively to solve problems.

**WinoGrande**: WinoGrande is a dataset of 44k problems for choosing the right option for a given sentence (Sakaguchi et al., 2021). It includes 40938 samples in the training set, 1,267 in the validation set, and 1,267 in the test set. The dataset is designed to assess models' commonsense reasoning abilities. The

examples contain sentences with fill-in-blanks that require the model to select the most appropriate option to complete the sentence.

**SocialIQA**: The SocialIQA dataset is a benchmark designed to evaluate a model's ability to reason about social interactions, including understanding social dynamics, intentions, and the effects of human actions (Sap et al., 2019). SocialIQA includes 33410 samples in the training set and 1954 in the validation set.

**PIQA**: The PIQA (Physical Interaction Question Answering) dataset is a benchmark designed to evaluate a model's ability to understand and reason about everyday physical interactions and affordances (Bisk et al., 2020). Here are some key details about PIQA: (Sakaguchi et al., 2021). PIQA contains 16113 samples in the training set and 1838 in the validation set.

