# OpenReview forum: "An Efficient Sparse Fine-Tuning with Low Quantization Error via Neural Network Pruning"
_TMLR — Accepted by TMLR_

### Review · Reviewer_11jE · 2025-08-19

**Summary Of Contributions:**

The paper proposes a Structured Pruning-Driven Sparse Fine-Tuning (SPruFT) framework, which selects a small subset of row/column parameters for fine-tuning based on neuron importance scores and employs low-rank decomposition for efficient training. The method is simple and implementation-friendly, with comprehensive experiments. Results on LLaMA-2/3 and multiple vision models outperform or match LoRA and other SOTA methods, while significantly reducing memory usage. Overall, the paper makes clear contributions with sufficient empirical evidence, though its novelty and depth of discussion remain limited.

Strengths:

1. No need for customized sparse operators, as it only relies on PyTorch’s native matrix multiplication for deployment.

2. Covers 8 commonsense reasoning benchmarks, GSM8k, MT-Bench, and 3 vision datasets, comparing against 10+ PEFT methods including LoRA, QLoRA, DoRA, and RoSA.

3. Derives the variance upper bound of Zeroth-Order Taylor importance estimation and provides a sampling complexity formula, laying a theoretical foundation for future research.

Weaknesses:

1. The core idea of “subnetwork selection via pruning importance + low-rank fine-tuning” is conceptually not new, resembling a combination of early Lottery Ticket Sparsity and LoRA.

2. Reports only a few sparsity ratios, without showing the performance degradation curve as sparsity increases.

3. The largest model evaluated is LLaMA-3-8B with sequence length ≤ 2048, leaving 70B-scale or longer-context scenarios unverified.

4. Some key symbols (e.g., η, M) lack intuitive illustrations when first introduced.

**Audience:**

Yes

**Audience Explanation:**

SPruFT provides a systematic solution to the critical challenge of efficiently fine-tuning billion-parameter models under limited GPU memory. The experiments span both LLaMA-2/3 and various vision Transformers, showing performance on par with LoRA and other state-of-the-art methods while further reducing memory usage by 20–50%. The method requires no custom CUDA kernels, and the accompanying code and scripts are open-sourced for reproducibility.

**Claims And Evidence:**

Yes

**Claims Explanation:**

The authors present systematic comparisons against more than ten PEFT baselines (including LoRA, QLoRA, DoRA, and RoSA) across a diverse set of benchmarks (8 commonsense reasoning tasks, GSM8k, MT-Bench, and 3 vision datasets). They further strengthen the empirical credibility of their work by providing detailed memory usage measurements, training hyperparameters, raw logs, and open-source code for full reproducibility.

**Requested Changes:**

The paper should provide the full sparsity–performance curve, as it currently reports results only at around 2% sparsity. I recommend adding experiments in the appendix covering the 0.5%–10% range (under the same task and rank), to clearly illustrate the degradation point and suggest a recommended sparsity interval.

Furthermore, the paper needs to more explicitly differentiate SPruFT from earlier approaches such as Lottery Ticket + LoRA, S-LoRA, or Block-Sparse LoRA. For instance, clarifying its novelty in terms of the neuron-importance objective, row–column selection strategy, or quantized-residual initialization would strengthen the contribution. Adding a dedicated comparison paragraph in the Related Work section would help address concerns about conceptual overlap.

---

### Review · Reviewer_JHRG · 2025-08-31

**Summary Of Contributions:**

In this paper, the author introduces SPruFT, a structured sparse fine-tuning method that selects the top-r neurons per layer and only trains their associated rows, reducing activation memory without custom kernels. Extensions include QSPruFT for quantized models using residual initialization, and ZOTaylor for efficient importance estimation with variance bounds. Experiments on LLaMA-2 and LLaMA-3 show competitive accuracy with LoRA-like methods while lowering memory.

Strengths:
- This paper tackles an important and prompt problem.
- Simple, kernel-agnostic design with consistent memory savings and competitive accuracy.
- Quantization-aware initialization that performs well in NF4.

Weaknesses:
- Some comparison between the proposed method and baselines seem not apple-to-apple. Your method “does not rely on dropout,” but others use dropout=0.1; please equalize dropout (and learning rates) to ensure memory/accuracy fairness. Also, for hardware consistency, some baselines ran on H100 vs A100; memory and throughput aren’t strictly comparable.
- Experiments are restricted to LLaMA-2 and LLaMA-3, both in the 7B–8B range. Missing broader validation on other popular open-source LLMs (e.g., Mistral, Falcon, Qwen) and larger-scale models makes it harder to generalize the claimed efficiency benefits.

**Audience:**

Yes

**Audience Explanation:**

Efficient fine-tuning of large models is an interesting, emerging, and important domain. The paper provides a simple, kernel-free alternative to LoRA-style approaches with both practical insight (graph-driven memory) and quantization integration, which would be of clear relevance to both researchers and practitioners.

**Broader Impact Concerns:**

No major ethical concerns beyond standard fine-tuning issues. The broader impact statement appears sufficient.

**Claims And Evidence:**

Yes

**Claims Explanation:**

The core methodology is correct and the empirical results support the claim that simpler computation graphs reduce memory even with fewer parameters. The residual initialization and zeroth-order scoring show clear benefits. However, fairness issues (dropout, learning rate, GPU differences) and small-scale training data reduce the strength of the validation. Theoretical claims for ZOTaylor are not fully aligned with practice but the empirical evidence is consistent.

**Requested Changes:**

Critical:

- Re-run or report baselines with matched dropout and learning rate for fair comparison.

- Clarify GPU differences in memory reporting (A100 vs H100).

Important but not critical:

- Provide variance/CI across multiple seeds given small training sets.

- Add one larger-data experiment for commonsense or GSM8K.

- Strengthen ZOTaylor discussion by clarifying theory–practice limits and adding nk sensitivity analysis.

- Extend evaluation beyond LLaMA-2/3 to include at least one other popular open-source model (e.g., Mistral, Falcon, Qwen) or larger-size LLaMAs.

Nice to have:

- Report wall-clock time and tokens/sec.

- Include GPT-4(o) MT-Bench scores for comparability.

---

> ### Author Response · Authors · 2025-09-19
> **Response to Reviewer JHRG**
>
> We are grateful to the reviewer for the thoughtful feedback and constructive suggestions.
>
> 1. The learning rate discrepancy was a writing mistake, and we apologize for the confusion. While our initial settings explored different learning rates (due to the absence of a scale coefficient in our method compared to LoRA), all reported results were obtained with a learning rate of $1×10^{-4}$. We will correct the description in the paper accordingly.
> 2. Our method is a sparse fine-tuning (SpFT) framework, which is itself a form of sparse training. Prior studies (see SpFT references in Section 2) have shown that SpFT is generally less prone to overfitting without dropout, which is one of its advantages. In contrast, introducing dropout into SpFT may sometimes lead to over-regularization. On the other hand, LoRA and its variants often benefit from dropout, as it may help reduce overfitting. Moreover, previous SpFT studies cited in our paper have also compared SpFT without dropout to LoRA variants with dropout. This motivated our design choice in the reported experiments. To verify this claim, we conduct a new experiment to examine the influence of dropout. Due to the limitation of characters, we will provide the comparisons in another response.
> 3. The memory usage difference for DoRA and RoSA between H100 and A100 is about 0.04GB (A100 being slightly lower). We ran these experiments on H100s because the total memory usage is close to the maximum available on a single A100.
> 4. As indicated in our reply to Reviewer 11jE, we face practical limitations in computational resources (e.g., job-level time and GPU allocation caps). Accordingly, while we recognize the value of the ``Important but not critical`` and ``Nice to have`` suggestions, we may not be able to conduct all of them in time. \
> 4.1. That said, we have the standard deviation of results on GSM8K on hand, which is generally around 1, indicating very small variability. This likely suggests that the sample size of 2048 is sufficiently large. Furthermore, across multiple datasets, our method consistently achieves performance comparable to other state-of-the-art approaches, which provides some evidence of generalizability. Finally, since we did not perform repeated experiments across multiple seeds for all tasks, we only claim that structured sparse fine-tuning can achieve results comparable to existing methods, rather than claiming that our method outperforms previous works.
> \begin{array}{|l|c|c|c|} \hline
> \text{GSM8k} & \text{Llama2(7B)} & \text{Llama3(8B)} \\\\\hline
> \text{LoRA}, r=64 & 18.42_{\pm1.068} & 41.77_{\pm1.358} \\\\
> \text{VeRA}, r=64 & 0.0_{\pm0.0} & 0.0_{\pm0.0} \\\\
> \text{RoSA}, r=32, d=0.12 & 21.99_{\pm1.141} & 45.72_{\pm1.372} \\\\
> \text{DoRA}, r=64 & 21.46_{\pm1.131} & 46.02_{\pm1.373} \\\\
> \text{SPruFT}, r=128 & 22.90_{\pm1.157} & 46.10_{\pm1.373} \\\\\hline
> \text{QLoRA}, r=64 & 18.88_{\pm1.078} & 41.17_{\pm1.356} \\\\
> \text{QDoRA}, r=64 & 25.63_{\pm1.203} & 46.17_{\pm1.373} \\\\
> \text{LoftQ}, r=64 & 20.55_{\pm1.113} & 44.96_{\pm1.370} \\\\
> \text{QPiSSA}, r=64 & 21.08_{\pm1.123} & 45.87_{\pm1.373} \\\\
> \text{QSPruFT}^*, r=128 & 20.32_{\pm1.108} & 47.76_{\pm1.376} \\\\
> \text{QSPruFT}^\dag, r=128 & 22.14_{\pm1.144} & 49.13_{\pm1.377} \\\\\hline\end{array}
> 4.2. ZOTaylor discussion: Generally, a larger value of $nk$ leads to more accurate estimates of importance scores. This increases the likelihood of capturing truly important neurons, which in turn may result in better fine-tuned models. However, the relationship between a neuron’s functional role and its empirical importance remains theoretically underexplored, and there is currently no reliable standard to guarantee what proportion of important neurons should be identified. Based on our experiments, we can only recommend increasing $nk$ as much as computational resources allow, in order to reduce the variance of the estimation.

---

> ### Author Response · Authors · 2025-09-19
> **Long tables provided here:**
>
> The following table reports the memory overhead introduced by adding a dropout layer: approximately 6.18 GB on LLaMA-2 and 6.35 GB on LLaMA-3. \
> \begin{array}{|l|c|c|c|} \hline
> \text{Model, ft setting} & \text{mem} \\\\ \hline
> \text{Llama2(7B)} & \\\\
> \text{SPruFT,} r=128, \text{w/ dropout} & 23.81\text{GB} \\\\
> \text{SPruFT,} r=128, \text{w/o dropout} & 17.62\text{GB} \\\\
> \text{LoRA,} r=64, \text{w/ dropout} & 23.46\text{GB} \\\\
> \text{LoRA,} r=128, \text{w/ dropout} & 25.98\text{GB} \\\\  \hline
> \text{Llama3(8B)} & \\\\
> \text{SPruFT,} r=128, \text{w/ dropout} & 30.84\text{GB} \\\\
> \text{SPruFT,} r=128, \text{w/o dropout} & 24.49\text{GB} \\\\
> \text{LoRA,} r=64, \text{w/ dropout} & 30.37\text{GB} \\\\
> \text{LoRA,} r=128, \text{w/ dropout} & 32.35\text{GB} \\\\ \hline\end{array}
>
> The comparison applying dropout to our approach. Sparsity: [$0.25$% ($r=16$), $8$% ($r=512$)].
>
> \begin{array}{|l|cc|cc|cc|cc|cc|cc|} \hline
> \text{Ours} & r=16 &  & r=32 &  & r=64 &  & r=128 &  & r=256 &  & r=512 &  \\\\
>  & \text{common(avg)} & \text{gsm8k} & \text{common(avg)} & \text{gsm8k} & \text{common(avg)} & \text{gsm8k} & \text{common(avg)} & \text{gsm8k} & \text{common(avg)} & \text{gsm8k} & \text{common(avg)} & \text{gsm8k} \\\\\hline
> \text{Llama2(7B)} & \\\\
> \text{random, w/ dropout} & 66.70 & 14.71 & 69.53 & 16.45 & 68.75 & 19.94 & 68.03 & 21.61 & 69.20 & 20.32 & 65.00 & 14.48 \\\\
> \text{random, w/o dropout} & 66.75 & 15.85 & 69.80 & 18.35 & 69.45 & 20.39 & 69.80 & 22.52 & 69.23 & 18.88 & 68.60 & 17.06 \\\\ \hline
> \ell^2, \text{w/ dropout} & 67.00 & 15.09 & 70.50 & 17.66 & 70.60 & 21.53 & 72.83 & 22.59 & 73.20 & 19.03 & 73.60 & 17.59 \\\\
> \ell^2, \text{w/o dropout} & 67.83 & 15.31 & 72.25 & 18.27 & 72.65 & 21.00 & 74.53 & 22.90 & 73.68 & 20.02 & 73.93 & 17.74 \\\\
> \hline
> \text{ZOTaylor, w/ dropout} & 68.38 & 15.77 & 71.80 & 23.43 & 70.28 & 21.00 & 73.83 & 22.59 & 74.63 & 22.21 & 73.23 & 17.13 \\\\
> \text{ZOTaylor, w/o dropout} & 70.70 & 17.82 & 72.35 & 23.81 & 72.80 & 22.06 & 74.35 & 23.50 & 73.95 & 24.94 & 73.45 & 16.07 \\\\
> \hline
> \text{Llama3(8B)} & \\\\
> \text{random, w/ dropout} & 76.50 & 32.15 & 76.35 & 40.03 & 73.78 & 46.17 & 76.55 & 43.29 & 74.80 & 37.45 & 74.10 & 33.13 \\\\
> \text{random, w/o dropout} & 78.00 & 35.03 & 77.75 & 36.54 & 75.18 & 42.23 & 76.08 & 42.53 & 76.95 & 38.29 & 75.68 & 36.62 \\\\ \hline
> \ell^2, \text{w/ dropout} & 73.53 & 31.46 & 77.75 & 43.29 & 77.03 & 43.75 & 79.78 & 45.79 & 77.70 & 36.77 & 77.93 & 34.72 \\\\
> \ell^2, \text{w/o dropout} & 75.43 & 33.59 & 78.53 & 41.70 & 77.88 & 46.25 & 80.55 & 46.10 & 78.80 & 38.89 & 78.70 & 37.45 \\\\
> \hline
> \text{ZOTaylor, w/ dropout} & 78.10 & 36.39 & 77.05 & 47.38 & 78.00 & 49.73 & 79.73 & 52.08 & 79.25 & 49.96 & 79.18 & 42.15 \\\\
> \text{ZOTaylor, w/o dropout} & 78.33 & 37.53 & 78.83 & 44.50 & 78.00 & 50.19 & 81.28 & 55.12 & 80.20 & 50.34 & 79.98 & 47.54 \\\\
> \hline
> \end{array}
>
> The comparison applying dropout to LoRA. Sparsity: [$0.25$% ($r=8$), $8$% ($r=256$)].
>
> \begin{array}{|l|cc|cc|cc|cc|cc|cc|} \hline
> \text{LoRA} & r=8 &  & r=16 &  & r=32 &  & r=64 &  & r=128 &  & r=256 & \\\\
>  & \text{common(avg)} & \text{gsm8k} & \text{common(avg)} & \text{gsm8k} & \text{common(avg)} & \text{gsm8k} & \text{common(avg)} & \text{gsm8k} & \text{common(avg)} & \text{gsm8k} & \text{common(avg)} & \text{gsm8k} \\\\\hline
> \text{Llama2(7B)} & \\\\
> \text{w/ dropout} & 73.55 & 15.39 & 72.48 & 19.86 & 72.03 & 19.56 & 74.23 & 18.42 & 74.53 & 22.90 & 73.90 & 18.20 \\\\
> \text{w/o dropout} & 72.63 & 15.16 & 72.98 & 19.64 & 72.70 & 17.82 & 73.15 & 18.88 & 72.78 & 21.30 & 73.65 & 17.36 \\\\ \hline
> \text{Llama3(8B)} & \\\\
> \text{w/ dropout} & 80.88 & 35.25 & 78.95 & 42.15 & 79.50 & 45.56 & 76.50 & 41.77 & 80.18 & 42.23 & 77.73 & 38.82 \\\\
> \text{w/o dropout} & 77.80 & 35.10 & 80.80 & 38.82 & 78.23 & 40.79 & 79.63 & 44.05 & 78.30 & 40.56 & 75.73 & 35.71 \\\\ \hline
> \end{array}
>
> We can observe that when the sparsity ratio is very small (0.25%), our approach performs less effectively, and LoRA achieves slightly better results. As the sparsity ratio increases, performance improvements diminish, suggesting that the range [0.5%, 2%] may be the most suitable. Somewhat surprisingly, performance decreases when the rank $r$ is large (256 for LoRA and 512 for our approach). We attribute this to the relatively small size of the fine-tuning dataset compared with the model size and the pre-training corpus. With more trainable parameters, the model is more likely to overfit. This observation is also consistent with the fact that many prior works using LoRA select rank values between 32 and 128, further supporting the view that PEFT can sometimes outperform full fine-tuning.
>
> Regarding the dropout layer, we see that SpFT generally does not rely on dropout, whereas LoRA appears to benefit slightly more from its use. In fact, applying dropout in our method provides little to no improvement and may even slightly degrade performance, while in LoRA it typically leads to performance gains.

---

### Review · Reviewer_x2ke · 2025-09-08

**Summary Of Contributions:**

# Summary Of Contributions
This paper introduces a novel Sparse Fine-Tuning (SpFT) framework inspired by neural network pruning techniques to enhance the memory and computational efficiency of fine-tuning large language models. The proposed method identifies important neurons using structural pruning-based feature importance metrics and confines fine-tuning to weights associated with these selected neurons. This targeted approach enables a substantial reduction in memory usage—achieving 20–50% savings—while maintaining competitive performance with state-of-the-art methods such as LoRA and its variants on standard language tasks.


# Strengths And Weaknesses

## Strengths
* The paper introduces a novel sparse fine-tuning (SpFT) framework that significantly improves memory efficiency over existing methods like LoRA, while relying solely on standard tensor operations, thus avoiding the complexities of custom sparse kernel implementations.

* It offers practical support for fine-tuning quantized models, extending its utility to resource-constrained settings and further reducing memory overhead.

* The authors present a thoughtful analysis of memory usage in PEFT methods, arguing that computation graphs play a more critical role in memory consumption than merely the number of trainable parameters—a useful insight for both researchers and practitioners.

* The work includes rigorous empirical validation across diverse fine-tuning tasks, coupled with actionable training strategies, enhancing its relevance and applicability.

* Two novel variants of Taylor importance—class-aware Taylor and Zeroth-Order Taylor—are proposed for different modalities. These demonstrate thoughtful adaptation to the specific needs of image classification and language modeling, respectively, with the latter requiring only a single forward pass in large models and featuring a variance reduction technique.


## Weakness

* The reliance on neural network pruning raises concerns about potential sensitivity to the pruning criteria; the robustness of the selected "important" neurons across different datasets or domains is not clearly discussed.

* The theoretical grounding or comparative advantages of the Taylor importance variants (class-aware and Zeroth-Order Taylor) over existing importance metrics remain under-explored. Overall, the principles behind choosing these two methods needs further exploration and discussions.

* There’s limited discussion on the potential training instability or convergence challenges introduced by the proposed novel work.

* The computation and memory performance gain under various model scales or tasks is not discussed. Moreover, Although the method avoids custom sparse kernels, it's unclear whether the standard tensor operations used can fully leverage existing hardware acceleration, especially at scale—potentially limiting the claimed practical efficiency.

* The integration of quantization is briefly mentioned, but the trade-offs between accuracy and compression in this context are not detailed.

* Empirical results on the image domain are on very small datasets. Are they conclusive enough in terms of the efficiency gained while training converged with stability while performance is maintained?

**Audience:**

Yes

**Audience Explanation:**

The paper very well motivates the problem behind the memory efficiency in addition to the parameter efficiency. The novel idea of introducing structured pruning methods in similar fashion to LoRA is very interesting. Therefore, it is very likely the audience would find the paper interesting and the findings motivational to further explore this direction in the future.

**Broader Impact Concerns:**

There are no "Broader Impact Concerns" for this paper.

**Claims And Evidence:**

No

**Claims Explanation:**

The claims are very well described and elaborated in the paper. The evidence is clear and accurate and well motivated and elaborated. There are some angles discussed in the weaknesses and requested changes sections regarding the convincing aspect of the paper that needs to be further discussed in the following reviewing stages. This will be revisited at the end for a final evaluation.

**Requested Changes:**

Based on the aforementioned strengths and weaknesses, the requested changes are provided in an orderly manner from the highest to the lowest importance as follows:

- Analysis over the sensitivity of the hyper-parameter r is missing. It is very instightful to show how much the gained efficiency is relied on a proper choince of r.

- Analysis and discussion over the training instability and convergence challenges with respect to the amount of efficiency (computational and memory) gained should be provided. It is unclear the how tight is the choince of the current hyper-parameters with repect to the compression gained over the course of the training for a stable training.

- There is a notation adding some confusion to the context in the second paragraph of section 4.1 where it says " with the row selection matrix Mijj = 1 ...".

- what is p in Eq. 1?

- Elaboration on this part is helpful as it is very concise and short on an important point; "However, MeZO uses the trick of regenerating z dynamically using a random seed (of much smaller size than the model), thus eliminating the storage and ensuring memory usage that is equal to that of inference. We then justify the effectiveness of ZOTaylor in the following." It seems very vague at this state and needs more clarification of the context and elaboration of the material to help the audience gain further insights into this trick.

- Can we look at the results of larger and more challenging image datasets to support the evidence behind the first importance approach vigorously?

---

> ### Author Response · Authors · 2025-09-19
> **Response to Reviewer x2ke:**
>
> We appreciate the reviewer’s thoughtful feedback, which will help us improve the paper.
> 1. We will include an analysis of the sensitivity of the hyperparameter $r$ using commonsense reasoning tasks and GSM8k. As noted in our response to Reviewer 11jE, we will incorporate experiments covering a range of sparsity ratios, which correspond to $r \in [8, 256]$ for LoRA and $r \in [16, 512]$ for our approach. The corresponding results are reported in our response to Reviewer JHRG.
> 2. We thank the reviewer for suggesting the inclusion of experiments on training stability and convergence. In the new experiments with different values of the hyperparameter $r$, we did observe that our approach performs less effectively when the sparsity ratio is very small (around $0.25$), whereas LoRA performs slightly better under the same conditions. However, we did not find evidence of convergence issues. The training loss consistently decreased rapidly across all experiments: all approaches, including our approach, LoRA, RoSA, and DoRA, yielded average losses of approximately $0.15\pm1$, $0.10\pm1$, and $0.05\pm1$ at the first, second, and third epochs, respectively, across all values of $r$ and tasks. Based on these results, we believe that training for three epochs is sufficient for our approach under our experimental settings.
> 3. Regarding the notation $M_{i_j j} = 1$, our intention is simply to emphasize that the $i_j^{\text{th}}$ row corresponding to the $j^{\text{th}}$ column is equal to $1$, so we prefer to keep this notation unchanged, with a line of explanation.
> 4. For the symbol $p$ in Eq. 11, it is meant to represent probability. We adopted this choice because in the default notation of TMLR, $p()$ is used to denote a probability distribution over continuous variables, which we find suitable here. We are, however, open to revising it to an alternative such as Pr[] if the reviewers or editors consider it clearer.
> 5. For the statement ``However, MeZO uses the trick of regenerating $z$ dynamically using a random seed …``, we did not elaborate in detail since this is from prior work, but we are happy to clarify it here. Earlier ZO methods before MeZO generate and store $z$, which requires memory on the scale of the model’s parameters. In contrast, MeZO stores only a random seed and regenerates the same $z$ values for each perturbation using a snippet like ``torch.manual_seed(random_seed); z = torch.normal(...)``. As the memory cost of storing a random seed is negligible compared to the parameters, the memory footprint of MeZO can essentially be regarded as comparable to inference.
> 6. Regarding the results on larger and more challenging image datasets, we selected the current benchmarks primarily because the underlying vision backbones are pretrained on ImageNet-1k, making these downstream tasks a natural choice. Our main purpose in including vision experiments was to enrich the evaluation, rather than to focus on scaling up vision models. In contrast, large language models involve substantially more parameters and pose a more critical challenge in terms of memory efficiency during training. That said, we agree that larger image datasets could provide additional insights. As a future direction, it would be interesting to explore our method on larger-scale vision models as well as multimodal or large vision-language models.

---

### Author Response · Authors · 2025-09-22
**Revision Track**

All section and table indices refer to the revised version of the manuscript.
1. Section 1, Paragraph 2: Added citations (the papers we mentioned in response to reviewer 11jE) for SpFT at “Another class of PEFT methods is sparse fine-tuning (SpFT, …)”.
2. Section 2, Low-Rank Adaptation (LoRA): Added citation of S-LoRA at “numerous variants have been proposed…,e.g.,...”.
3. Section 2: Added a new paragraph titled “Differentiate Our Work From Prior Works” to clarify the distinctions between our approach and existing methods.
4. Section 4.1, Paragraph 2: Added the phrase “, where $M$ can be viewed as a mask” to provide intuition for the use of symbol $M$.
5. Section 4.3.2: Added the sentence “Intuitively, Theorem 4.4 indicates that …” after Theorem 4.4 to explain the usage of $nk$ and improve the coherence of the final paragraph.
6. Section 5, Paragraph 3: Corrected the description “use a learning rate of $10^{−4}$ with linear decay (rate 0.01) for all methods”
7. Added the section “Dropout Comparison and $r$ Sensitivity Analysis” along with related content. \
7.1. Section 6: Added section 6.3 “Dropout Comparison and $r$ Sensitivity Analysis”, Table 7, and Table 8. \
7.2. Appendix D.6, Table 16 and Table 17: Added results of $r=8,128, 256$ for LoRA and $r=16, 256, 512$ for our approach. \
7.3. Appendix D.6: Modifies “$r=16$ is sufficient for LoRA” to be “$r=8$ is sufficient for LoRA”. \
7.4. Appendix D.6 and Section 6, Footnote 8: Modifies “lower than that of LoRA with $r = 16$” to be “lower than that of LoRA with $r = 8$”.
8. Reordered Table 5 and Table 6 to improve layout and optimize space utilization.
9. Table 5 used “rd” and “ZO” to denote “random” and “ZOTaylor,” respectively, for consistency with Table 8. Abbreviations were adopted in Table 8 due to layout and space constraints.

---

### Decision · Action_Editor_kBHF · 2025-10-15

**Recommendation:** Accept as is

**Additional Comments:**

The 3 reviewers gave constructive feedback which were appropriately addressed by the authors in the discussion phase, after which all reviewers agree and recommend the submission to be accepted.

**Audience:**

Yes

**Audience Explanation:**

Yes, the submission is relevant to TMLR's audience.

**Claims And Evidence:**

Yes

**Claims Explanation:**

All reviewers agree that, after the discussion and revision, the claims made in the submission are supported by accurate, convincing and clear evidence

---

> ### Author Response · Authors · 2025-10-21
> **Thank you for handling our submission**
>
> Dear Guillaume and all reviewers,
>
> Thank you very much for handling our paper and for your thoughtful coordination throughout the review process.
> We sincerely appreciate your time and effort, as well as the constructive feedback from the reviewers, which helped us improve the paper.